# SPLIT LBI FOR DEEP LEARNING: STRUCTURAL SPARSITY VIA DIFFERENTIAL INCLUSION PATHS

## ABSTRACT

Over-parameterization is ubiquitous nowadays in training neural networks to benefit both optimization in seeking global optima and generalization in reducing prediction error. However, compressive networks are desired in many real world applications and direct training of small networks may be trapped in local optima. In this paper, instead of pruning or distilling over-parameterized models to compressive ones, we propose a new approach based on *differential inclusions of inverse scale spaces*, that generates a family of models from simple to complex ones by coupling gradient descent and mirror descent to explore model structural sparsity. It has a simple discretization, called the Split Linearized Bregman Iteration (SplitLBI), whose global convergence analysis in deep learning is established that from any initializations, algorithmic iterations converge to a critical point of empirical risks. Experimental evidence shows that SplitLBI may achieve comparable and even better performance than other training algorithms on ResNet-18 in large scale training on ImageNet-2012 dataset *etc.*, while with *early stopping* it unveils effective subnet architecture with comparable test accuracy to dense models after retraining instead of pruning well-trained ones.

## 1 INTRODUCTION

The expressive power of deep neural networks comes from the millions of parameters, which are optimized by Stochastic Gradient Descent (SGD) (Bottou, 2010) and variants like Adam (Kingma & Ba, 2015). Remarkably, model over-parameterization helps both optimization and generalization. For optimization, over-parameterization may simplify the landscape of empirical risks toward locating global optima efficiently by gradient descent method (Mei et al., 2018; 2019; Venturi et al., 2018; Allen-Zhu et al., 2018; Du et al., 2018). On the other hand, over-parameterization does not necessarily result in a bad generalization or overfitting (Zhang et al., 2017), especially when some weight-size dependent complexities are controlled (Bartlett, 1997; Bartlett et al., 2017; Golowich et al., 2018; Neyshabur et al., 2019).

However, compressive networks are desired in many real world applications, *e.g.* robotics, self-driving cars, and augmented reality. Despite that $\ell_1$ regularization has been applied to deep learning to enforce the sparsity on weights toward compact, memory efficient networks, it sacrifices some prediction performance (Collins & Kohli, 2014). This is because that the weights learned in neural networks are highly correlated, and $\ell_1$ regularization on such weights violates the incoherence or irrepresentable conditions needed for sparse model selection (Donoho & Huo, 2001; Tropp, 2004; Zhao & Yu, 2006), leading to spurious selections with poor generalization. On the other hand, $\ell_2$ regularization is often utilized for correlated weights as some low-pass filtering, sometimes in the form of weight decay (Loshchilov & Hutter, 2019) or early stopping (Yao et al., 2007; Wei et al., 2017). Furthermore, group sparsity regularization (Yuan & Lin, 2006) has also been applied to neural networks, such as finding optimal number of neuron groups (Alvarez & Salzmann, 2016) and exerting good data locality with structured sparsity (Wen et al., 2016; Yoon & Hwang, 2017).

Yet, without the aid of over-parameterization, directly training a compressive model architecture may meet the obstacle of being trapped in local optima in contemporary experience. Alternatively, researchers in practice typically start from training a big model using common task datasets like ImageNet, and then prune or distill such big models to small ones without sacrificing too much of the performance (Jaderberg et al., 2014; Han et al., 2015; Zhu et al., 2017; Zhou et al., 2017; Zhang et al.,

2016; Li et al., 2017; Abbasi-Asl & Yu, 2017; Yang et al., 2018; Arora et al., 2018). In particular, a recent study (Frankle & Carbin, 2019) created the *lottery ticket hypothesis* based on empirical observations: "dense, randomly-initialized, feed-forward networks contain subnetworks (winning tickets) that – when trained in isolation – reach test accuracy comparable to the original network in a similar number of iterations". How to effectively reduce an over-parameterized model thus becomes the key to compressive deep learning. Yet, Liu et al. (2019) raised a question, *is it necessary to fully train a dense, over-parameterized model before finding important structural sparsity*?

In this paper, we provide a novel answer by exploiting a dynamic approach to deep learning with structural sparsity. We are able to establish a family of neural networks, from simple to complex, by following regularization paths as solutions of *differential inclusions of inverse scale spaces*. Our key idea is to design some dynamics that simultaneously exploit over-parameterized models and structural sparsity. To achieve this goal, the original network parameters are lifted to a coupled pair, with one *weight set $W$* of parameters following the standard gradient descend to explore the over-parameterized model space, while the other set of parameters learning structure sparsity in an *inverse scale space*, *i.e.*, *structural sparsity set* $\Gamma$. The large-scale important parameters are learned at a fast speed while the small unimportant ones are learned at a slow speed. The two sets of parameters are coupled in an $\ell_2$ regularization. The dynamics enjoys a simple discretization, *i.e.* the Split Linearized Bregman Iteration (SplitLBI), with provable global convergence guarantee shown in this paper. Here, SplitLBI is a natural extension of SGD with structural sparsity exploration: SplitLBI reduces to the standard gradient descent method when the coupling regularization is weak, while it leads to a sparse mirror descent when the coupling is strong.

Critically, SplitLBI enjoys a nice property that important subnet architecture can be rapidly learned via the structural sparsity parameter $\Gamma$ following the iterative regularization path, without fully training a dense network first. Particularly, the support set of structural sparsity parameter $\Gamma$ learned in the early stage of this inverse scale space discloses important sparse subnet architectures. Such architectures can be fine-tuned or retrained to achieve comparable test accuracy as the dense, over-parameterized networks. As a result, the structural sparsity parameter $\Gamma$ may enable us to rapidly find "winning tickets" in early training epochs for the "lottery" of identifying successful subnetworks that bear comparable test accuracy to the dense ones. This point is empirically validated in our experiments.

Historically, the *Linearized Bregman Iteration (LBI)* was firstly proposed in applied mathematics as iterative regularization paths for image reconstruction and compressed sensing (Osher et al., 2005; Yin et al., 2008), later applied to logistic regression (Shi et al., 2013). The convergence analysis was given for convex problems (Yin et al., 2008; Cai et al., 2009), yet remaining open for non-convex problems met in deep learning. Osher et al. (2016) established statistical model selection consistency for high dimensional linear regression under the same irrepresentable condition as Lasso, later extended to generalized linear models (Huang & Yao, 2018). To relax such conditions, SplitLBI was proposed by Huang et al. (2016) to learn structural sparsity in linear models under weaker conditions than generalized Lasso, that was successfully applied in medical image analysis (Sun et al., 2017) and computer vision (Zhao et al., 2018). In this paper, it is the first time that SplitLBI is exploited to train highly non-convex neural networks with structural sparsity, together with a global convergence analysis based on the Kurdyka-Łojasiewicz framework Łojasiewicz (1963).

**Contributions**. (1) SplitLBI, as an extension of SGD, is applied to deep learning by exploring both over-parameterized models and structural sparsity in the inverse scale space. (2) Global convergence of SplitLBI in such a nonconvex optimization is established based on the Kurdyka-Łojasiewicz framework, that the whole iterative sequence converges to a critical point of the empirical loss function from arbitrary initializations. (3) Stochastic variants of SplitLBI demonstrate the comparable and even better performance than other training algorithms on ResNet-18 in large scale training such as ImageNet-2012, among other datasets, together with additional structural sparsity in successful models for interpretability. (4) Structural sparsity parameters in SplitLBI provide important information about subnetwork architecture with comparable or even better accuracies than dense models before and after retraining -- *SplitLBI with early stopping* can provide fast "*winning tickets*" without fully training dense, over-parameterized models.

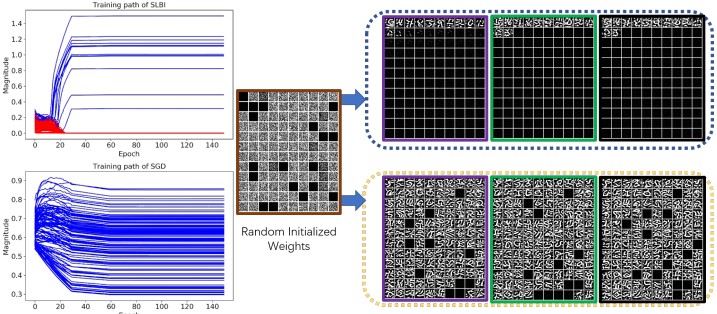

Figure 1: Visualization of solution path and filter patterns in the third convolutional layer (i.e., conv.c5) of LetNet-5, trained on MNIST. The left figure shows the magnitude changes for each filter of the models trained by SplitLBI and SGD, where $x$-axis and $y$-axis indicate the training epochs, and filter magnitudes ($\ell_2$-norm), respectively. The SplitLBI path of filters selected in the support of $\Gamma$ are drawn in blue color, while the red color curves represent the filters that are not important and outside the support of $\Gamma$. We visualize the corresponding learned filters by Erhan et al. (2009) at 20 (blue), 40 (green), and 80 (black) epochs, which are shown in the right figure with the corresponding color bounding boxes, *i.e.*, blue, green, and black, respectively. It shows that our SplitLBI enjoys a sparse selection of filters without sacrificing accuracy (see Table 1).

## 2 METHODOLOGY

The supervised learning task learns a mapping $\Phi_W : \mathcal{X} \to \mathcal{Y}$, from input space $\mathcal{X}$ to output space $\mathcal{Y}$, with a parameter $W$ such as weights in neural networks, by minimizing certain loss functions on training samples $\widehat{\mathcal{L}}_n(W) = \frac{1}{n} \sum_{i=1}^{n} \ell(y_i, \Phi_W(x_i))$. For example, a neural network of $l$-layer is defined as $\Phi_W(x) = \sigma_l(W^l \sigma_{l-1}(W^{l-1} \cdots \sigma_1(W^1 x))$, where $W = \{W^i\}_{i=1}^l$, $\sigma_i$ is the nonlinear activation function of the $i$-th layer.

**Differential Inclusion of Inverse Scale Space.** Consider the following dynamics,

$$\frac{\dot{W}_t}{\kappa} = -\nabla_W \bar{\mathcal{L}}(W_t, \Gamma_t) \tag{1a}$$

$$\dot{V}_t = -\nabla_\Gamma \bar{\mathcal{L}}(W_t, \Gamma_t) \tag{1b}$$

$$V_t \in \partial \bar{\Omega}(\Gamma_t) \tag{1c}$$

where $V$ is a sub-gradient of $\bar{\Omega}(\Gamma) := \Omega_\lambda(\Gamma) + \frac{1}{2\kappa} \|\Gamma\|^2$ for some sparsity-enforced regularization $\Omega_\lambda(\Gamma) = \lambda \Omega_1(\Gamma)$ ($\lambda \in \mathbb{R}_+$) such as Lasso or group Lasso penalties for $\Omega_1(\Gamma)$, $\kappa > 0$ is a damping parameter such that the solution path is continuous, and the augmented loss function is

$$\bar{\mathcal{L}}(W, \Gamma) = \widehat{\mathcal{L}}_n(W) + \frac{1}{2\nu} \|W - \Gamma\|_2^2, \tag{2}$$

with $\nu > 0$ controlling the gap admitted between $W$ and $\Gamma$. Compared to the original loss function $\widehat{\mathcal{L}}_n(W)$, the $\bar{\mathcal{L}}(W, \Gamma)$ additionally adopt the variable splitting strategy, by lifting the original neural network parameter $W$ to $(W, \Gamma)$ with $\Gamma$ modeling the structural sparsity of $W$. For simplicity, we assumed $\bar{\mathcal{L}}$ is differentiable with respect to $W$ here, otherwise the gradient in Eq. (1a) is understood as subgradient and the equation becomes an inclusion.

The differential inclusion system (1), called *Split Inverse Scale Space* (SplitISS), can be understood as a gradient descent flow of $W_t$ in the proximity of $\Gamma_t$ and a mirror descent flow (Nemirovski & Yudin, 1983) of $\Gamma_t$ associated with a sparsity enforcement penalty $\bar{\Omega}$. In mirror descent flow, gradient descent goes on the dual space consisting of sub-gradients $V_t$, driving the flow in sparse primal space of $\Gamma_t$. For a large enough $\nu$, it reduces to the gradient descent method for $W_t$. Yet the solution path of $\Gamma_t$ exhibits the following property in the separation of scales: starting at the zero, important parameters of large scale will be learned fast, popping up to be nonzeros early, while unimportant parameters of small scale will be learned slowly, appearing to be nonzeros late. In fact, taking $\Omega_\lambda(\Gamma) = \|\Gamma\|_1$ and $\kappa \to \infty$ for simplicity, $V_t$ as the subgradient of $\bar{\Omega}_t$, undergoes a gradient descent flow before reaching the $\ell_\infty$-unit box, which implies that $\Gamma_t = 0$ in this stage. The earlier a component in $V_t$ reaches

the $\ell_\infty$-unit box, the earlier a corresponding component in $\Gamma_t$ becomes nonzero and rapidly evolves toward a critical point of $\bar{\mathcal{L}}$ under gradient flow. On the other hand, the $W_t$ follows the gradient descent with a standard $\ell_2$-regularization. Therefore, $W_t$ closely follows dynamics of $\Gamma_t$ whose important parameters are selected. Such a property is called as the *inverse scale space* in applied mathematics (Burger et al., 2006) and recently was shown to achieve statistical model selection consistency in high dimensional linear regression (Osher et al., 2016) and general linear models (Huang & Yao, 2018), with a reduction of bias as $\kappa$ increases. In this paper, we shall see that the inverse scale space property still holds empirically for the highly nonconvex neural network training via Eq. (1). For example, Fig. 1 shows a LeNet trained on MNIST by the discretized dynamics, where important sparse filters are selected in early epochs while the popular SGD returns dense filters.

Compared with directly enforcing a penalty function such as $\ell_1$ or $\ell_2$ regularization

$$\min_W \widehat{\mathcal{R}}_n(W) := \widehat{\mathcal{L}}_n(W) + \Omega_\lambda(W), \quad \lambda \in \mathbb{R}_+. \tag{3}$$

SplitISS avoids the parameter correlation problem in over-parameterized models. In fact, a necessary and sufficient condition for Lasso or $\ell_1$-type sparse model selection is the incoherence or irrepresentable conditions (Tropp (2004); Zhao & Yu (2006)) that are violated for highly correlated weight parameters, leading to spurious discoveries. In contrast, Huang et al. (2018) showed that equipped with such a variable splitting where $\Gamma$ enjoys an orthogonal design where the restricted Hessian of the augmented loss on $\Gamma$ is orthogonal, the SplitISS can achieve model selection consistency under weaker conditions than generalized Lasso, relaxing the incoherence or irrepresentable conditions when parameters are highly correlated. For weight parameter $W$, instead of directly being imposed with $\ell_1$-sparsity, it adopts $\ell_2$-regularization in the proximity of the sparse path of $\Gamma$ that admits simultaneously exploring highly correlated parameters in over-parameterized models and sparsity regularization.

**Split Linearized Bregman Iterations.** SplitISS admits an extremely simple discrete approximation, using the Euler forward discretization of dynamics (1):

$$W_{k+1} = W_k - \kappa\alpha_k \cdot \nabla_W \bar{\mathcal{L}}(W_k, \Gamma_k), \tag{4a}$$

$$V_{k+1} = V_k - \alpha_k \cdot \nabla_\Gamma \bar{\mathcal{L}}(W_k, \Gamma_k), \tag{4b}$$

$$\Gamma_{k+1} = \kappa \cdot \mathrm{Prox}_{\Omega_\lambda}(V_{k+1}), \tag{4c}$$

where $V_0 = \Gamma_0 = 0$, $W_0$ can be small random numbers such as Gaussian distribution in neural networks, for some complex networks it can be initialized as common setting. The proximal map in Eq. (4c) that controls the sparsity of $\Gamma$ is given by

$$\mathrm{Prox}_{\Omega_\lambda}(V) = \arg\min_\Gamma \left\{ \frac{1}{2}\|\Gamma - V\|_2^2 + \Omega_\lambda(\Gamma) \right\}, \tag{5}$$

We shall call such an iterative procedure as *Split Linearized Bregman Iteration* (SplitLBI), that was firstly coined in Huang et al. (2016) as an iterative regularization path for sparse modeling in high dimensional statistics. In the application to neural networks, the loss becomes highly non-convex, the SplitLBI returns a sequence of sparse models from simple to complex ones whose global convergence condition to be shown below, while solving Eq. (3) at various levels of $\lambda$ might not be tractable except for over-parameterized models.

The sparsity-enforcement penalty used in convolutional neural networks can be chosen as follows. Our sparsity framework aims at regularizing the groups of weight parameters using group Lasso penalty (Yuan & Lin, 2006), $\Omega_1(\Gamma) = \sum_g \|\Gamma^g\|_2$, where $\|\Gamma^g\|_2 = \sqrt{\sum_{i=1}^{|\Gamma^g|}(\Gamma_i^g)^2}$ and $|\Gamma^g|$ is the number of weights in $\Gamma^g$. Thus Eq. (4c) has a closed form solution $\Gamma^g = \kappa \cdot \max(0, 1 - 1/\|V^g\|_2) V^g$ for the $g$-th filter. We treat convolutional and fully connected layers in different ways.

(1) For a convolutional layer, $\Gamma^g = \Gamma^g(c_{in}, c_{out}, \texttt{size})$ denote the convolutional filters where `size` denotes the kernel size and $c_{in}$ and $c_{out}$ denote the numbers of input channels and output channels, respectively. When we regard each group as each convolutional filter, $g = c_{out}$; otherwise for weight sparsity, $g$ can be every element in the filter that reduces to the Lasso.

(2) For a fully connected layer, $\Gamma = \Gamma(c_{in}, c_{out})$ where $c_{in}$ and $c_{out}$ denote the numbers of inputs and outputs of the fully connected layer. Each group $g$ corresponds to each element $(i, j)$, and the group Lasso penalty degenerates to the Lasso penalty.

## 3 GLOBAL CONVERGENCE OF SPLITLBI FOR NEURAL NETWORKS

We present a theorem that guarantees the *global convergence* of SplitLBI, *i.e.* from any intialization, the SplitLBI sequence converges to a critical point of $\bar{\mathcal{L}}$. Our treatment extends the block coordinate descent (BCD) studied in Zeng et al. (2019), with a crucial difference being the mirror descent involved in SplitLBI. Instead of the splitting loss in BCD (Zeng et al., 2019), a new Lyapunov function is developed here to meet the Kurdyka-Łojasiewicz property Łojasiewicz (1963). Xue & Xin (2018) studied convergence of variable splitting method for single hidden layer networks with Gaussian inputs.

Let $P := (W, \Gamma)$. Following Huang & Yao (2018), the SplitLBI algorithm in Eq. (4a-4c) can be rewritten as the following standard Linearized Bregman Iteration,

$$P_{k+1} = \arg\min_{P} \left\{ \langle P - P_k, \alpha \nabla \bar{\mathcal{L}}(P_k) \rangle + B_{\Psi}^{p_k}(P, P_k) \right\}, \tag{6}$$

where

$$\Psi(P) = \Omega_\lambda(\Gamma) + \frac{1}{2\kappa}\|P\|_2^2 = \Omega_\lambda(\Gamma) + \frac{1}{2\kappa}\|W\|_2^2 + \frac{1}{2\kappa}\|\Gamma\|_2^2, \tag{7}$$

$p_k \in \partial\Psi(P_k)$, and $B_{\Psi}^q$ is the Bregman divergence associated with convex function $\Psi$, defined by

$$B_{\Psi}^q(P, Q) := \Psi(P) - \Psi(Q) - \langle q, P - Q \rangle, \text{ for some } q \in \partial\Psi(Q). \tag{8}$$

Without loss of generality, consider $\lambda = 1$ in the sequel. One can establish the global convergence of SplitLBI under the following assumptions.

**Assumption 1.** *Suppose that: (a)* $\widehat{\mathcal{L}}_n(W) = \frac{1}{n}\sum_{i=1}^{n}\ell(y_i, \Phi_W(x_i))$ *is continuous differentiable and* $\nabla\widehat{\mathcal{L}}_n$ *is Lipschitz continuous with a positive constant* $Lip$; *(b)* $\widehat{\mathcal{L}}_n(W)$ *has bounded level sets; (c)* $\widehat{\mathcal{L}}_n(W)$ *is lower bounded (without loss of generality, we assume that the lower bound is 0); (d)* $\Omega$ *is a proper lower semi-continuous convex function and has locally bounded subgradients, that is, for every compact set* $\mathcal{S} \subset \mathbb{R}^n$, *there exists a constant* $C > 0$ *such that for all* $\Gamma \in \mathcal{S}$ *and all* $g \in \partial\Omega(\Gamma)$, *there holds* $\|g\| \leq C$; *and (e) the Lyapunov function*

$$F(P, G) := \alpha\bar{\mathcal{L}}(W, \Gamma) + B_{\Omega}^{\tilde{g}}(\Gamma, \tilde{\Gamma}), \tag{9}$$

*is a Kurdyka-Łojasiewicz function on any bounded set, where* $B_{\Omega}^{\tilde{g}}(\Gamma, \tilde{\Gamma}) := \Omega(\Gamma) - \Omega(\tilde{\Gamma}) - \langle\tilde{g}, \Gamma - \tilde{\Gamma}\rangle$, $\tilde{\Gamma} \in \partial\Omega^*(\tilde{g})$, *and* $\Omega^*$ *is the conjugate of* $\Omega$ *defined as*

$$\Omega^*(g) := \sup_{U \in \mathbb{R}^n} \{\langle U, g \rangle - \Omega(U)\}.$$

**Remark 1.** *Assumption 1 (a)-(c) are regular in the analysis of nonconvex algorithm (see, Attouch et al. (2013) for instance), while Assumption 1 (d) is also mild including all Lipschitz continuous convex function over a compact set. Some typical examples satisfying Assumption 1(d) are the* $\ell_1$ *norm, group* $\ell_1$ *norm, and every continuously differentiable penalties. By Eq. (9) and the definition of conjugate, the Lyapunov function* $F$ *can be rewritten as follows,*

$$F(W, \Gamma, g) = \alpha\bar{\mathcal{L}}(W, \Gamma) + \Omega(\Gamma) + \Omega^*(g) - \langle\Gamma, g\rangle. \tag{10}$$

Now we are ready to present the main theorem.

**Theorem 1.** *[Global Convergence of SplitLBI] Suppose that Assumption 1 holds. Let* $(W_k, \Gamma_k)$ *be the sequence generated by SplitLBI (Eq. (4a-4c)) with a finite initialization. If*

$$0 < \alpha_k = \alpha < \frac{2}{\kappa(Lip + \nu^{-1})},$$

*then* $(W_k, \Gamma_k)$ *converges to a critical point of* $\bar{\mathcal{L}}$ *defined in Eq. (2), and* $\{W^k\}$ *converges to a critical point of* $\widehat{\mathcal{L}}_n(W)$.

Applying to the neural networks, typical examples are summarized in the following corollary.

**Corollary 1.** *Let* $\{W_k, \Gamma_k, g_k\}$ *be a sequence generated by SLBI ([16a-16c](#)) for neural network training where (a)* $\ell$ *is any smooth definable loss function, such as the square loss* $(t^2)$*, exponential loss* $(e^t)$*, logistic loss* $\log(1 + e^{-t})$*, and cross-entropy loss; (b)* $\sigma_i$ *is any smooth definable activation, such as linear activation* $(t)$*, sigmoid* $(\frac{1}{1+e^{-t}})$*, hyperbolic tangent* $(\frac{e^t - e^{-t}}{e^t + e^{-t}})$*, and softplus* $(\frac{1}{c} \log(1 + e^{ct})$ *for some* $c > 0$*) as a smooth approximation of ReLU; (c)* $\Omega$ *is the group Lasso. Then the sequence* $\{W_k\}$ *converges to a stationary point of* $\widehat{\mathcal{L}}_n(W)$ *under the conditions of Theorem 1.*

Proofs of Theorem 1 and Corollary 1 are given in Appendix A.

## 4 EXPERIMENTS WITH STOCHASTIC SPLITLBI

We begin with some stochastic variants of SplitLBI and implementations, followed by four groups of experiments demonstrating the utilities of weight parameter $W_t$ and structural sparsity parameter $\Gamma_t$ in prediction, interpretability, and capturing effective sparse subnetworks.

**Batch Split LBI**. For neural network training with large datasets, stochastic approximation of the gradients in Split LBI over the mini-batch $(\mathbf{X}, \mathbf{Y})_{\text{batch}_t}$ is adopted to update the parameter $W$,

$$\widetilde{\nabla}_W^t = \nabla_W \widehat{\mathcal{L}}_n(W) \mid_{(\mathbf{X}, \mathbf{Y})_{\text{batch}_t}}. \tag{11}$$

**SplitLBI with momentum (Mom)**. Inspired by the variants of SGD, the momentum term can be also incorporated to the standard Split LBI that leads to the following updates of $W$ by replacing Eq ([4a](#)) with,

$$v_{t+1} = \tau v_t + \widetilde{\nabla}_W \bar{\mathcal{L}}(W_t, \Gamma_t) \tag{12a}$$
$$W_{t+1} = W_t - \kappa \alpha v_{t+1} \tag{12b}$$

where $\tau$ is the momentum factor, empirically setting as 0.9 in default. One immediate application of such stochastic algorithms of SplitLBI is to "boost networks", *i.e.* growing a network from the null to a complex one by sequentially applying our algorithm on subnets with increasing complexities.

**SplitLBI with momentum and weight decay (Mom-Wd).** The update formulation is,

$$v_{t+1} = \tau v_t + \widetilde{\nabla}_W \bar{\mathcal{L}}(W_t, \Gamma_t) \tag{13}$$
$$W_{t+1} = W_t - \kappa \alpha v_{t+1} - \beta W_t \tag{14}$$

where $\beta$ is set as $1e^{-4}$.

**Implementation.** Various algorithms are evaluated over the various backbones – LeNet ([LeCun et al.](#), [2015](#)), AlexNet ([Krizhevsky et al.](#), [2012](#)), VGG ([Simonyan & Zisserman](#), [2014](#)), and ResNet ([He et al.](#), [2016](#)) etc., respectively. For MNIST and Cifar-10, the default hyper-parameters of Split LBI are $\kappa = 1$, $\nu = 10$ and $\alpha_k$ is set as 0.1, decreased by 1/10 every 30 epochs. In ImageNet-2012, the Split LBI utilizes $\kappa = 1$, $\nu = 1000$, and $\alpha_k$ is initially set as 0.1, decays 1/10 every 30 epochs. We set $\lambda = 1$ in Eq. ([5](#)) by default, unless otherwise specified. On MNIST and Cifar-10, the batch size is set as 128; and for all methods, the batch size of ImageNet 2012 is 256. The standard data augmentation implemented in pytorch is applied to Cifar-10 and ImageNet2012 datasets, as [He et al.](#) ([2016](#)). The weights of all models are initialized as [He et al.](#) ([2015](#)). In the following experiments, we define *sparsity* as percentage of non-zero parameters, i.e. the number of non-zero weights dividing the total number of weights in consideration, that equals to one minus the pruning rate of the network. We also have the reproducible source codes [1].

### 4.1 IMAGE CLASSIFICATION

In SplitLBI, the weight parameter $W_t$ explores over-parameterized models that can achieve the state-of-the-art performance in large scale training such as ImageNet-2012 classification.

**Experimental Design**. We compare different variants of SGD and Adam in the experiments. By default, the learning rate of competitors is set as 0.1 for SGD and its variant and 0.001 for Adam and its variants, and gradually decreased by 1/10 every 30 epochs. In particular, we have,

---

[1] https://anonymous.4open.science/repository/d22bbbc8-50d5-4e60-b2e8-4ded4e93db63/Split_LBI_code

| Dataset | | MNIST | Cifar-10 | ImageNet-2012 | |
| --- | --- | --- | --- | --- | --- |
| Models | Variants | LeNet | ResNet-20 | AlexNet | ResNet-18 |
| *SGD* | Naive | 98.87 | 86.46 | –/– | 60.76/79.18 |
| | $l_1$ | 98.52 | 67.60 | –/– | –/– |
| | Mom | 99.16 | 89.44 | 55.14/78.09 | 66.98/86.97 |
| | Mom-W$d^\star$ | **99.23** | **90.31** | 56.55/79.09 | 69.76/89.18 |
| | Nesterov | **99.23** | 90.18 | -/- | 70.19/89.30 |
| *Adam* | Naive | 99.19 | 89.14 | –/– | 59.66/83.28 |
| | Adabound | 99.15 | 87.89 | –/– | –/– |
| | Adagrad | 99.02 | 88.17 | –/– | –/– |
| | Amsgrad | 99.14 | 88.68 | –/– | –/– |
| | Radam | 99.08 | 88.44 | –/– | –/– |
| *SplitLBI* | Naive | 99.02 | 89.26 | 55.06/77.69 | 65.26/86.57 |
| | Mom | 99.19 | 89.72 | 56.23/78.48 | 68.55/87.85 |
| | Mom-Wd | 99.20 | 89.95 | **57.09/79.86** | **70.55/89.56** |

Table 1: Top-1/Top-5 accuracy(%) on ImageNet-2012 and test accuracy on MNIST/Cifar-10. $^\star$: results from the official pytorch website. We use the official pytorch codes to run the competitors. All models are trained by 100 epochs. In this table, we run the experiment by ourselves except for SGD Mom-Wd on ImageNet which is reported in https://pytorch.org/docs/stable/torchvision/models.html.

SGD: (1) Naive SGD: the standard SGD with batch input. (2) SGD with $l_1$ penalty (Lasso). The $l_1$ norm is applied to penalize the weights of SGD by encouraging the sparsity of learned model, with the regularization parameter of the $l_1$ penalty term being set as $1e^{-3}$ (3) SGD with momentum (Mom): we utilize momentum 0.9 in SGD. (4) SGD with momentum and weight decay (Mom-Wd): we set the momentum 0.9 and the standard $l_2$ weight decay with the coefficient weight $1e^{-4}$. (5) SGD with Nesterov (Nesterov): the SGD uses nesterov momentum 0.9.

Adam: (1) Naive Adam: it refers to the standard version of Adam. We report the results of several recent variants of Adam, including (2) Adabound, (3) Adagrad, (4) Amsgrad, and (5) Radam.

**SplitLBI achieves the state-of-the-art performance on ImageNet-2012, etc.** Tab. 1 shows the experimental results on ImageNet-2012, Cifar-10, and MNIST of some classical networks -- LeNet, AlexNet and ResNet. Our SplitLBI variants may achieve comparable or even better performance than SGD variants in 100 epochs, indicating the efficacy in learning dense, over-parameterized models.

## 4.2 SPLITLBI LEARNS SPARSE FILTERS FOR IMPROVED INTERPRETATION

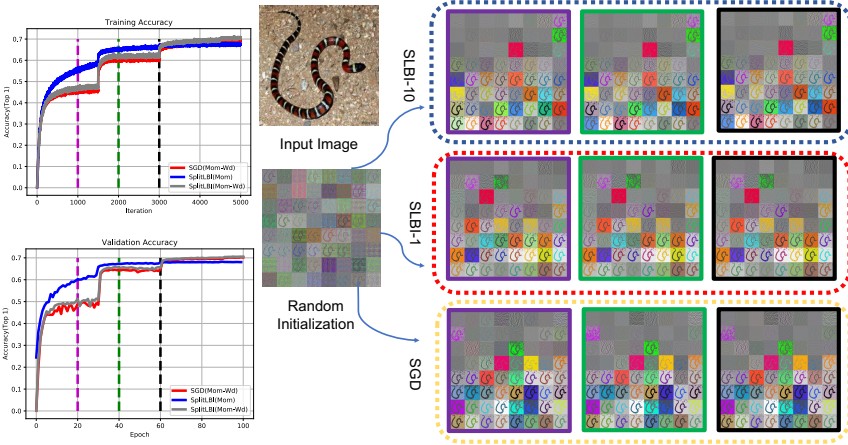

Figure 2: Visualization of the first convolutional layer filters of ResNet-18 trained on ImageNet-2012. Given the input image and initial weights visualized in the middle, filter response gradients at 20 (purple), 40 (green), and 60 (black) epochs are visualized by Springenberg et al. (2014). The "SLBI-10" ("SLBI-1") in the right figure refers to SplitLBI with $\kappa = 10$ and $\kappa = 1$, respectively.

In SplitLBI, the structural sparsity parameter $\Gamma_t$ explores important sub-network architectures that contributes significantly to the loss or error reduction in early training stages. Through the $\ell_2$-coupling, structural sparsity parameter $\Gamma_t$ may guide the weight parameter to explore those sparse models in favour of improved interpretabiity. For example, Fig. 1 visualizes some sparse filters learned by SplitLBI of LeNet-5 trained on MNIST (with $\kappa = 10$ and weight decay every 40 epochs), in comparison with dense filters learned by SGD. The activation pattern of such sparse filters favours high order global correlations between pixels of input images. To further reveal the insights of learned patterns of SplitLBI, we visualize the first convolutional layer of ResNet-18 on ImageNet-2012 along the training path of our SplitLBI as in Fig. 2. The left figure compares the training and validation accuracy of SplitLBI and SGD. The right figure compares visualizations of the filters learned by SplitLBI and SGD using Springenberg et al. (2014).

**Implementation.** To be specific, denote the weights of an $l$-layer network as $\{W^1, W^2, \cdots, W^l\}$. For the $i-$th layer weights $W^i$, denote the $j-$th channel $W^i_j$. Then we compute the gradient of the sum of the feature map computed from each filter $W^i_j$ with respect to the input image (here a snake image). We further conduct the min-max normalization to the gradient image, and generate the final visualization map. The right figure compares the visualized gradient images of first convolutional layer of 64 filters with $7 \times 7$ receptive fields. We visualize the models parameters at 20 (purple), 40 (green), and 60 (black) epochs, respectively, which corresponds to the bounding boxes in the right figure annotated by the corresponding colors, *i.e.*, purple, green, and black. We order the gradient images produced from 64 filters by the descending order of the magnitude ($\ell_2$-norm) of filters, *i.e.*, images are ordered from the upper left to the bottom right. For comparison, we also provide the visualized gradient from random initialized weights.

**Filters learned by ImageNet prefer to non-semantic texture rather than shape and color.** The filters of high norms mostly focus on the texture and shape information, while color information is with the filters of small magnitudes. This phenomenon is in accordance with observation of Abbasi-Asl & Yu (2017) that filters mainly of color information can be pruned for saving computational cost. Moreover, among the filters of high magnitudes, most of them capture non-semantic textures while few pursue shapes. This shows that the first convolutional layer of ResNet-18 trained on ImageNet learned non-semantic textures rather than shape to do image classification tasks, in accordance with recent studies (Geirhos et al., 2019). How to enhance the semantic shape invariance learning, is arguably a key to improve the robustness of convolutional neural networks.

### 4.3 GLOBAL CONVERGENCE AND STRUCTURAL SPARSITY OF SPLITLBI

We conduct ablation studies based on Cifar-10 dataset with VGG-16 and ResNet-56 to evaluate (i) global convergence of $\bar{\mathcal{L}}$; and (ii) the structural sparsity learned by $\Gamma_t$ via exploring test accuracies of sparse models obtained by projecting $W_t$ onto the support set of $\Gamma_t$ (mask), by varying two key hyper-parameters $\kappa$ and $\nu$.

**Implementation.** We choose SplitLBI with momentum and weight decay, since it achieves very good performance on large-scale experiments. Specifically, we have two set of experiments, where each experiment is repeated for 5 times: (1) we fix $\nu = 100$ and vary $\kappa = 1, 2, 5, 10$, where sparsity of $\Gamma_t$ and validation accuracies of sparse models are shown in top row of Fig. 4. Note that we keep $\kappa \cdot \alpha_k = 0.1$ in Eq (1a), to make comparable learning rate of each variant, and also consistent with SGD. Thus the learning rate $\alpha_k$ will be adjusted by different $\kappa$ values. (2) we fix $\kappa = 1$, and validate the results of SplitLBI with $\nu = 10, 20, 50, 100, 200, 500, 1000, 2000$ in the second row of Fig. 4 with the learning rate $\alpha_k = 0.1$. Moreover, rather than using sparse models associated with $\Gamma_t$, Fig. 6 in Appendix shows the validation accuracies of full models learned by $W_t$.

**SplitLBI converges to Critical Point**. Figure 3 shows the curves of training loss $(\widehat{\mathcal{L}}_n)$ and accuracies, with each point representing the average and variance bar over 5 times. As shown, both training loss and training accuracy will converge, which validates our theoretical result in Theorem 1. Besides, larger $\kappa$ brings in slower convergence, which agrees with the analysis the convergence rate is inversely scale to $\kappa$ in Lemma A.5.

**Sparse subnetworks achieve comparable performance to dense models without fine-tuning or retraining**. From the experiments above, the sparsity of $\Gamma$ grows as $\kappa$ and $\nu$ increase. While large $\kappa$ may cause a small number of important parameters growing rapidly, large $\nu$ will decouple $W_t$ and $\Gamma_t$ such that the growth of $W_t$ does not affect $\Gamma_t$ that may over-sparsify and deteriorate model accuracies.

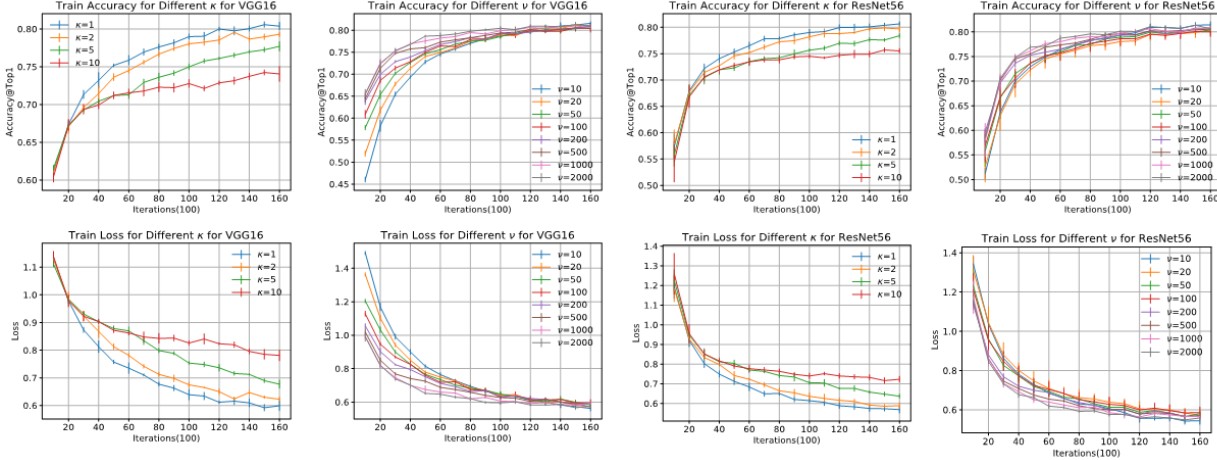

Figure 3: Loss and training accuracy by different $\kappa$ and $\nu$ validates the global convergence of SplitLBI. The X-axis and Y-axis indicate the training epochs, and sparsity/accuracy. The results are repeated for 5 rounds, by keeping the exactly same initialization for each model. In each round, we use the same initialization for every hyperparameter. For all models, we train for 160 epochs with initial learning rate (lr) of 0.1 and decrease by 0.1 at epoch 80 and 120.

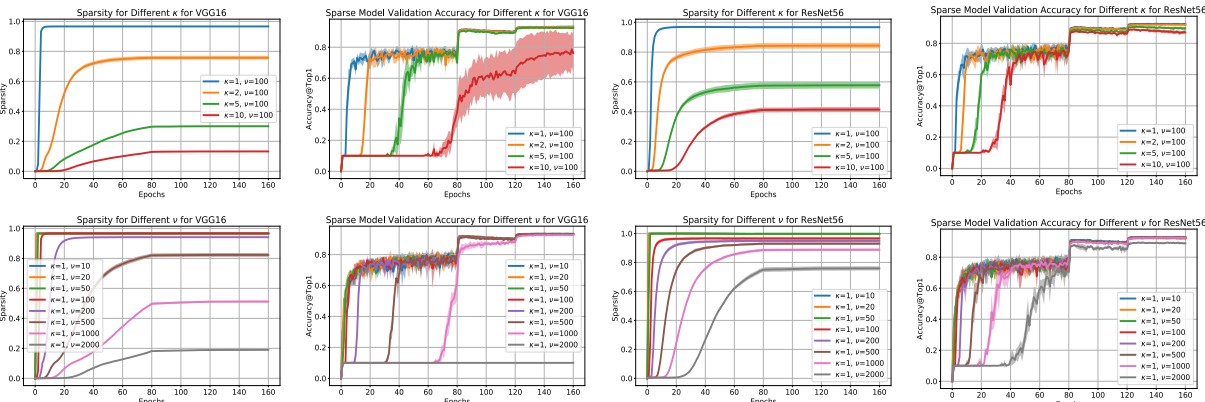

Figure 4: Sparsity and validation accuracy by different $\kappa$ and $\nu$ show that moderate sparse models may achieve comparable test accuracies to dense models without fine-tuning. Sparsity is obtained as the percentage of nonzeros in $\Gamma_t$ and sparse model at epoch $t$ is obtained by projection of $W_t$ onto the support set of $\Gamma_t$, i.e. pruning the weights corresponding to zeros in $\Gamma_t$. The best accuracies achieved are recorded in Tab. 2 and 4 of Appendix for different $\kappa$ and $\nu$, respectively. X-axis and Y-axis indicate the training epochs, and sparsity/accuracy. The results are repeated for 5 times. Shaded area indicates the variance; and in each round, we keep the exactly same initialization for each model. In each round, we use the same initialization for every hyperparameter. For all the model, we train for 160 epochs with initial learning rate (lr) of 0.1 and decrease by 0.1 at epoch 80 and 120.

Thus a moderate choice of $\kappa$ and $\nu$ is preferred in practice. In all cases, one can see that moderate sparse models can achieve comparable predictive power to dense models, even without fine-tuning or retraining. This shows that the structural sparsity parameter $\Gamma_t$ can indeed capture important weight parameter $W_t$ through their coupling.

## 4.4 SPLITLBI WITH EARLY STOPPING AND RETRAIN FINDS EFFECTIVE SUB-NETWORKS

Equipped with early stopping, $\Gamma_t$ in early epochs may learn effective subnetworks (*i.e.* "winning tickets" (Frankle & Carbin, 2019; Liu et al., 2019)) that achieve comparable or even better performance after retraining than existing pruning strategies by SGD.

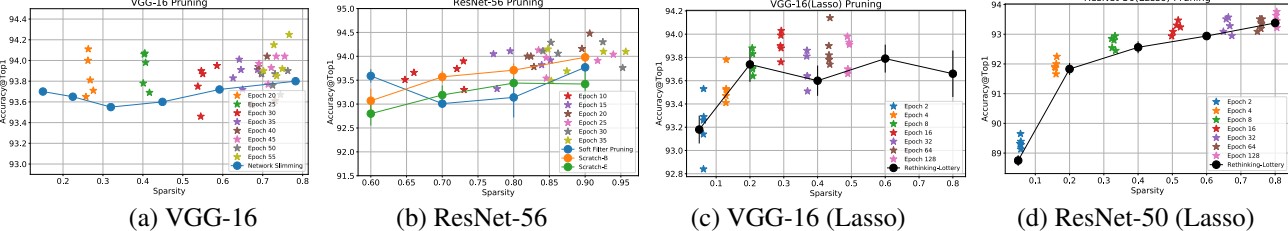

Figure 5: SplitLBI with early stopping finds sparse subnets whose test accuracies (stars) after retrain are comparable or even better than the baselines (Network Slimming (reproduced by the released codes from Liu et al. (2019) ) , Soft-Filter Pruning(Tab. 10), Scratch-B(Tab. 10), Scratch-E(Tab. 10), and "Rethinking-Lottery" (Tab. 9a)) as reported in Liu et al. (2019). Sparse filters of VGG-16 and ResNet-56 are show in (a) and (b), while sparse weights of VGG-16 and ResNet-50 are shown in (c) and (d).

**Experimental Design.** We adopt a comparison baseline as the one-shot pruning strategy in Frankle & Carbin (2019), which firstly trains a dense over-parameterized model by SGD for $T = 160$ epochs and find the sparse structure by pruning weights or filters (Liu et al., 2019), then secondly retrains the structure from the scratch with $T$ epochs from the same initialization as the first step. For SplitLBI, instead of pruning weights/filters from dense models, we directly utilize the structural sparsity $\Gamma_t$ at different training epochs to define the subnet architecture, followed by retrain-from-scratch (Fine-tune is shown in Appendix Sec. D with preliminary results). Experiments are conducted on Cifar-10 dataset where we still use VGG–16, ResNet-50, and ResNet-56 as the networks to make direct comparisons to previous works. SplitLBI uses momentum and weight decay with hyperparameters shown in Tab. 10 in Appendix. In particular, we set $\lambda = 0.1$, and $0.05$ for VGG-16, and ResNet-56 respectively, since ResNet-56 has less parameters than VGG-16. Furthermore, we introduce another variant of our SplitLBI by using Lasso rather than group lasso penalty for $\Gamma_t$ to sparsify the weights of convolutional filters; and the corresponding models are denoted as VGG-16 (Lasso) and ResNet-50 (Lasso). Every experiment is repeated for five times and the results are shown in Fig. 5. Note that in different runs of SplitLBI, the sparsity of $\Gamma_t$ slightly varies.

**Sparse subnets found by early stopping of SplitLBI achieve remarkably good accuracy after retrain from scratch.** In Fig.5 (a-b), sparse filters discovered by $\Gamma_t$ at different epochs are compared against the methods of Network Slimming (Liu et al., 2017), Soft Filter Pruning (Yang et al., 2018), Scratch-B, and Scratch-E, whose results are reported from Liu et al. (2019). At similar sparsity levels, SplitLBI can achieve comparable or even better accuracy than competitors, even with sparse architecture learned from very early epochs (e.g. $t = 20$ or $10$). Moreover in Fig.5 (c-d), we can draw the same conclusion for the sparse weights of VGG-16 (Lasso) and ResNet-50 (Lasso), against the results reported in Liu et al. (2019). These results shows that the structural sparsity parameter $\Gamma_t$ found by early stopping of SplitLBI already discloses important subnetwork architecture that may achieve remarkably good accuracy after retrain from scratch. Therefore, it is not necessary to fully train a dense model to find a successful sparse subnet architecture with comparable performance to the dense ones -- one can early stop SplitLBI properly where the structural parameter $\Gamma_t$ unveils "*winning tickets*" (Frankle & Carbin, 2019).

## 5 CONCLUSION

In this paper, a parsimonious deep learning method is proposed based on differential inclusions of inverse scale spaces. Implemented by a variable splitting scheme, such a dynamics system can exploit over-parameterized models and structural sparsity simultaneously. Besides, its simple discretization, i.e., the SplitLBI, has a proven global convergence and hence can be employed to train deep networks. We have experimentally shown that it can achieve the state-of-the-art performance on many datasets including ImageNet-2012, with better interpretability than SGD. What's more, equipped with early stopping, such a structural sparsity can unveil the "winning tickets" – the architecture of sub-networks which after re-training can achieve comparable and even better accuracy than original dense networks.

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

APPENDIX TO *SplitLBI for deep learning: structural sparsity via differential inclusion paths*

## A  PROOF OF THEOREM 1

First of all, we reformulate Eq. (6) into an equivalent form. Without loss of generality, consider $\Omega = \Omega_1$ in the sequel.

Denote $R(P) := \Omega(\Gamma)$, then Eq. (6) can be rewritten as,

$$P_{k+1} = \text{Prox}_{\kappa R}(P_k + \kappa(p_k - \alpha\nabla\bar{\mathcal{L}}(P_k))), \tag{15a}$$

$$p_{k+1} = p_k - \kappa^{-1}(P_{k+1} - P_k + \kappa\alpha\nabla\bar{\mathcal{L}}(P_k)), \tag{15b}$$

where $p_k = [0, g_k]^T \in \partial R(P_k)$ and $g_k \in \partial\Omega(\Gamma_k)$. Thus SplitLBI is equivalent to the following iterations,

$$W_{k+1} = W_k - \kappa\alpha\nabla_W\bar{\mathcal{L}}(W_k, \Gamma_k), \tag{16a}$$

$$\Gamma_{k+1} = \text{Prox}_{\kappa\Omega}(\Gamma_k + \kappa(g_k - \alpha\nabla_\Gamma\bar{\mathcal{L}}(W_k, \Gamma_k))), \tag{16b}$$

$$g_{k+1} = g_k - \kappa^{-1}(\Gamma_{k+1} - \Gamma_k + \kappa\alpha \cdot \nabla_\Gamma\bar{\mathcal{L}}(W_k, \Gamma_k)). \tag{16c}$$

Exploiting the equivalent reformulation (16a-16c), one can establish the global convergence of $(W_k, \Gamma_k, g_k)$ based on the Kurdyka-Łojasiewicz framework. In this section, the following extended version of Theorem 1 is actually proved.

**Theorem 2.** *[Global Convergence of SplitLBI] Suppose that Assumption 1 holds. Let $(W_k, \Gamma_k, g_k)$ be the sequence generated by SplitLBI (Eq. (16a-16c)) with a finite initialization. If*

$$0 < \alpha_k = \alpha < \frac{2}{\kappa(Lip + \nu^{-1})},$$

*then $(W_k, \Gamma_k, g_k)$ converges to a critical point of $F$. Moreover, $\{(W_k, \Gamma_k)\}$ converges to a stationary point of $\bar{\mathcal{L}}$ defined in Eq. 2, and $\{W^k\}$ converges to a stationary point of $\widehat{\mathcal{L}}_n(W)$.*

### A.1  KURDYKA-ŁOJASIEWICZ PROPERTY

To introduce the definition of the Kurdyka-Łojasiewicz (KL) property, we need some notions and notations from variational analysis, which can be found in Rockafellar & Wets (1998).

The notion of subdifferential plays a central role in the following definitions. For each $\mathbf{x} \in \text{dom}(h) := \{\mathbf{x} \in \mathbb{R}^p : h(\mathbf{x}) < +\infty\}$, the *Fréchet subdifferential* of $h$ at $\mathbf{x}$, written $\widehat{\partial}h(\mathbf{x})$, is the set of vectors $\mathbf{v} \in \mathbb{R}^p$ which satisfy

$$\lim_{\mathbf{y}\neq\mathbf{x},\mathbf{y}\to\mathbf{x}} \inf \frac{h(\mathbf{y}) - h(\mathbf{x}) - \langle\mathbf{v}, \mathbf{y} - \mathbf{x}\rangle}{\|\mathbf{x} - \mathbf{y}\|} \geq 0.$$

When $\mathbf{x} \notin \text{dom}(h)$, we set $\widehat{\partial}h(\mathbf{x}) = \varnothing$. The *limiting-subdifferential* (or simply *subdifferential*) of $h$ introduced in Mordukhovich (2006), written $\partial h(\mathbf{x})$ at $\mathbf{x} \in \text{dom}(h)$, is defined by

$$\partial h(\mathbf{x}) := \{\mathbf{v} \in \mathbb{R}^p : \exists \mathbf{x}^k \to \mathbf{x}, \ h(\mathbf{x}^k) \to h(\mathbf{x}), \ \mathbf{v}^k \in \widehat{\partial}h(\mathbf{x}^k) \to \mathbf{v}\}. \tag{17}$$

A necessary (but not sufficient) condition for $\mathbf{x} \in \mathbb{R}^p$ to be a minimizer of $h$ is $\mathbf{0} \in \partial h(\mathbf{x})$. A point that satisfies this inclusion is called *limiting-critical* or simply *critical*. The distance between a point $\mathbf{x}$ to a subset $\mathcal{S}$ of $\mathbb{R}^p$, written $\text{dist}(\mathbf{x}, \mathcal{S})$, is defined by $\text{dist}(\mathbf{x}, \mathcal{S}) = \inf\{\|\mathbf{x} - \mathbf{s}\| : \mathbf{s} \in \mathcal{S}\}$, where $\|\cdot\|$ represents the Euclidean norm.

Let $h : \mathbb{R}^p \to \mathbb{R} \cup \{+\infty\}$ be an extended-real-valued function (respectively, $h : \mathbb{R}^p \rightrightarrows \mathbb{R}^q$ be a point-to-set mapping), its *graph* is defined by

$$\text{Graph}(h) := \{(\mathbf{x}, y) \in \mathbb{R}^p \times \mathbb{R} : y = h(\mathbf{x})\},$$

$$(\text{resp. } \text{Graph}(h) := \{(\mathbf{x}, \mathbf{y}) \in \mathbb{R}^p \times \mathbb{R}^q : \mathbf{y} \in h(\mathbf{x})\}),$$

and its domain by $\mathrm{dom}(h) := \{\mathbf{x} \in \mathbb{R}^p : h(\mathbf{x}) < +\infty\}$ (resp. $\mathrm{dom}(h) := \{\mathbf{x} \in \mathbb{R}^p : h(\mathbf{x}) \neq \varnothing\}$). When $h$ is a proper function, i.e., when $\mathrm{dom}(h) \neq \varnothing$, the set of its global minimizers (possibly empty) is denoted by

$$\arg\min h := \{\mathbf{x} \in \mathbb{R}^p : h(\mathbf{x}) = \inf h\}.$$

The KL property (Łojasiewicz, 1963; 1993; Kurdyka, 1998; Bolte et al., 2007a;b) plays a central role in the convergence analysis of nonconvex algorithms (Attouch et al., 2013; Wang et al., 2019). The following definition is adopted from Bolte et al. (2007b).

**Definition 1.** *[Kurdyka-Łojasiewicz property] A function $h$ is said to have the Kurdyka-Łojasiewicz (KL) property at $\bar{u} \in \mathrm{dom}(\partial h) := \{v \in \mathbb{R}^n | \partial h(v) \neq \emptyset\}$, if there exists a constant $\eta \in (0, \infty)$, a neighborhood $\mathcal{N}$ of $\bar{u}$ and a function $\phi : [0, \eta) \to \mathbb{R}_+$, which is a concave function that is continuous at $0$ and satisfies $\phi(0) = 0$, $\phi \in \mathcal{C}^1((0, \eta))$, i.e., $\phi$ is continuous differentiable on $(0, \eta)$, and $\phi'(s) > 0$ for all $s \in (0, \eta)$, such that for all $u \in \mathcal{N} \cap \{u \in \mathbb{R}^n | h(\bar{u}) < h(u) < h(\bar{u}) + \eta\}$, the following inequality holds*

$$\phi'(h(u) - h(\bar{u})) \cdot \mathrm{dist}(0, \partial h(u)) \geq 1. \tag{18}$$

*If $h$ satisfies the KL property at each point of $\mathrm{dom}(\partial h)$, $h$ is called a KL function.*

KL functions include real analytic functions, semialgebraic functions, tame functions defined in some o-minimal structures (Kurdyka, 1998; Bolte et al., 2007b), continuous subanalytic functions (Bolte et al., 2007a) and locally strongly convex functions. In the following, we provide some important examples that satisfy the Kurdyka-Łojasiewicz property.

**Definition 2.** *[Real analytic] A function $h$ with domain an open set $U \subset \mathbb{R}$ and range the set of either all real or complex numbers, is said to be **real analytic** at $u$ if the function $h$ may be represented by a convergent power series on some interval of positive radius centered at $u$: $h(x) = \sum_{j=0}^{\infty} \alpha_j (x - u)^j$, for some $\{\alpha_j\} \subset \mathbb{R}$. The function is said to be **real analytic** on $V \subset U$ if it is real analytic at each $u \in V$ (Krantz & Parks, 2002, Definition 1.1.5). The real analytic function $f$ over $\mathbb{R}^p$ for some positive integer $p > 1$ can be defined similarly.*

*According to Krantz & Parks (2002), typical real analytic functions include polynomials, exponential functions, and the logarithm, trigonometric and power functions on any open set of their domains. One can verify whether a multivariable real function $h(\mathbf{x})$ on $\mathbb{R}^p$ is analytic by checking the analyticity of $g(t) := h(\mathbf{x} + t\mathbf{y})$ for any $\mathbf{x}, \mathbf{y} \in \mathbb{R}^p$.*

**Definition 3.** *[Semialgebraic]*

*(a) A set $\mathcal{D} \subset \mathbb{R}^p$ is called semialgebraic (Bochnak et al., 1998) if it can be represented as*

$$\mathcal{D} = \bigcup_{i=1}^{s} \bigcap_{j=1}^{t} \{\mathbf{x} \in \mathbb{R}^p : P_{ij}(\mathbf{x}) = 0, Q_{ij}(\mathbf{x}) > 0\},$$

*where $P_{ij}, Q_{ij}$ are real polynomial functions for $1 \leq i \leq s, 1 \leq j \leq t$.*

*(b) A function $h : \mathbb{R}^p \to \mathbb{R} \cup \{+\infty\}$ (resp. a point-to-set mapping $h : \mathbb{R}^p \rightrightarrows \mathbb{R}^q$) is called semialgebraic if its graph $\mathrm{Graph}(h)$ is semialgebraic.*

According to (Łojasiewicz, 1965; Bochnak et al., 1998) and (Shiota, 1997, I.2.9, page 52), the class of semialgebraic sets are stable under the operation of finite union, finite intersection, Cartesian product or complementation. Some typical examples include polynomial functions, the indicator function of a semialgebraic set, and the Euclidean norm (Bochnak et al., 1998, page 26).

## A.2 KL PROPERTY IN DEEP LEARNING AND PROOF OF COROLLARY 1

In the following, we consider the deep neural network training problem. Consider a $l$-layer feedforward neural network including $l - 1$ hidden layers of the neural network. Particularly, let $d_i$ be the number of hidden units in the $i$-th hidden layer for $i = 1, \ldots, l - 1$. Let $d_0$ and $d_l$ be the number of units of input and output layers, respectively. Let $W^i \in \mathbb{R}^{d_i \times d_{i-1}}$ be the weight matrix between the $(i - 1)$-th layer and the $i$-th layer for any $i = 1, \ldots l$[2].

---

[2]To simplify notations, we regard the input and output layers as the $0$-th and the $l$-th layers, respectively, and absorb the bias of each layer into $W^i$.

According to Theorem 2, one major condition is to verify the introduced Lyapunov function $F$ defined in (9) satisfies the Kurdyka-Łojasiewicz property. For this purpose, we need an extension of semialgebraic set, called the *o-minimal structure* (see, for instance Coste (1999), van den Dries (1986), Kurdyka (1998), Bolte et al. (2007b)). The following definition is from Bolte et al. (2007b).

**Definition 4.** *[o-minimal structure] An o-minimal structure on $(\mathbb{R}, +, \cdot)$ is a sequence of boolean algebras $\mathcal{O}_n$ of "definable" subsets of $\mathbb{R}^n$, such that for each $n \in \mathbb{N}$*

  *(i) if $A$ belongs to $\mathcal{O}_n$, then $A \times \mathbb{R}$ and $\mathbb{R} \times A$ belong to $\mathcal{O}_{n+1}$;*

  *(ii) if $\Pi : \mathbb{R}^{n+1} \to \mathbb{R}^n$ is the canonical projection onto $\mathbb{R}^n$, then for any $A$ in $\mathcal{O}_{n+1}$, the set $\Pi(A)$ belongs to $\mathcal{O}_n$;*

  *(iii) $\mathcal{O}_n$ contains the family of algebraic subsets of $\mathbb{R}^n$, that is, every set of the form*
  $$\{x \in \mathbb{R}^n : p(x) = 0\},$$
  *where $p : \mathbb{R}^n \to \mathbb{R}$ is a polynomial function.*

  *(iv) the elements of $\mathcal{O}_1$ are exactly finite unions of intervals and points.*

Based on the definition of o-minimal structure, we can show the definition of the *definable function*.

**Definition 5.** *[Definable function] Given an o-minimal structure $\mathcal{O}$ (over $(\mathbb{R}, +, \cdot)$), a function $f : \mathbb{R}^n \to \mathbb{R}$ is said to be definable in $\mathcal{O}$ if its graph belongs to $\mathcal{O}_{n+1}$.*

According to van den Dries & Miller (1996); Bolte et al. (2007b), there are some important facts of the o-minimal structure, shown as follows.

  (i) The collection of *semialgebraic* sets is an o-minimal structure. Recall the semialgebraic sets are Bollean combinations of sets of the form
  $$\{x \in \mathbb{R}^n : p(x) = 0, q_1(x) < 0, \ldots, q_m(x) < 0\},$$
  where $p$ and $q_i$'s are polynomial functions in $\mathbb{R}^n$.

  (ii) There exists an o-minimal structure that contains the sets of the form
  $$\{(x, t) \in [-1, 1]^n \times \mathbb{R} : f(x) = t\}$$
  where $f$ is real-analytic around $[-1, 1]^n$.

  (iii) There exists an o-minimal structure that contains simultaneously the graph of the exponential function $\mathbb{R} \ni x \mapsto \exp(x)$ and all semialgebraic sets.

  (iv) The o-minimal structure is stable under the sum, composition, the inf-convolution and several other classical operations of analysis.

The Kurdyka-Łojasiewicz property for the smooth definable function and non-smooth definable function were established in (Kurdyka, 1998, Theorem 1) and (Bolte et al., 2007b, Theorem 11), respectively. Now we are ready to present the proof of Corollary 1.

*Proof.* [Proof of Corollary 1] To justify this corollary, we only need to verify the associated Lyapunov function $F$ satisfies Kurdyka-Łojasiewicz inequality. In this case and by (10), $F$ can be rewritten as follows

$$F(\mathcal{W}, \Gamma, \mathcal{G}) = \alpha \left( \widehat{\mathcal{L}}_n(W, \Gamma) + \frac{1}{2\nu} \|W - \Gamma\|^2 \right) + \Omega(\Gamma) + \Omega^*(g) - \langle W, g \rangle.$$

Because $\ell$ and $\sigma_i$'s are definable by assumptions, then $\widehat{\mathcal{L}}_n(W, \Gamma)$ are definable as compositions of definable functions. Moreover, according to Krantz & Parks (2002), $\|W - \Gamma\|^2$ and $\langle W, g \rangle$ are semi-algebraic and thus definable. Since the group Lasso $\Omega(\Gamma) = \sum_g \|\Gamma\|_2$ is the composition of $\ell_2$ and $\ell_1$ norms, and the conjugate of group Lasso penalty is the maximum of group $\ell_2$-norm, *i.e.* $\Omega^*(\Gamma) = \max_g \|\Gamma_g\|_2$, where the $\ell_2$, $\ell_1$, and $\ell_\infty$ norms are definable, hence the group Lasso and its conjugate are definable as compositions of definable functions. Therefore, $F$ is definable and hence satisfies Kurdyka-Łojasiewicz inequality by (Kurdyka, 1998, Theorem 1).

The verifications of other cases listed in assumptions can be found in the proof of (Zeng et al., 2019, Proposition 1). This finishes the proof of this corollary. $\qquad\square$

### A.3 PROOF OF THEOREM 2

Our analysis is mainly motivated by a recent paper (Benning et al., 2017), as well as the influential work (Attouch et al., 2013). According to (Attouch et al., 2013, Lemma 2.6), there are mainly four ingredients in the analysis, that is, the *sufficient descent property*, *relative error property*, *continuity property* of the generated sequence and the *Kurdyka-Łojasiewicz property* of the function. More specifically, we first establish the *sufficient descent property* of the generated sequence via exploiting the Lyapunov function $F$ (see, (9)) in Lemma A.4 in Section A.4, and then show the *relative error property* of the sequence in Lemma A.5 in Section A.5. The *continuity property* is guaranteed by the continuity of $\bar{\mathcal{L}}(W, \Gamma)$ and the relation $\lim_{k\to\infty} B_\Omega^{g_k}(\Gamma_{k+1}, \Gamma_k) = 0$ established in Lemma 1(i) in Section A.4. Thus, together with the Kurdyka-Łojasiewicz assumption of $F$, we establish the global convergence of SLBI following by (Attouch et al., 2013, Lemma 2.6).

Let $(\bar{W}, \bar{\Gamma}, \bar{g})$ be a critical point of $F$, then the following holds

$$
\begin{aligned}
\partial_W F(\bar{W}, \bar{\Gamma}, \bar{g}) &= \alpha(\nabla\widehat{\mathcal{L}}_n(\bar{W}) + \nu^{-1}(\bar{W} - \bar{\Gamma})) = 0, \\
\partial_\Gamma F(\bar{W}, \bar{\Gamma}, \bar{g}) &= \alpha\nu^{-1}(\bar{\Gamma} - \bar{W}) + \partial\Omega(\bar{\Gamma}) - \bar{g} \ni 0, \\
\partial_g F(\bar{W}, \bar{\Gamma}, \bar{g}) &= \bar{\Gamma} - \partial\Omega^*(\bar{g}) \ni 0.
\end{aligned}
\tag{19}
$$

By the final inclusion and the convexity of $\Omega$, it implies $\bar{g} \in \partial\Omega(\bar{\Gamma})$. Plugging this inclusion into the second inclusion yields $\alpha\nu^{-1}(\bar{\Gamma} - \bar{W}) = 0$. Together with the first equality implies

$$
\nabla\bar{\mathcal{L}}(\bar{W}, \bar{\Gamma}) = 0, \quad \nabla\widehat{\mathcal{L}}_n(\bar{W}) = 0.
$$

This finishes the proof of this theorem.

### A.4 SUFFICIENT DESCENT PROPERTY ALONG LYAPUNOV FUNCTION

Let $P_k := (W_k, \Gamma_k)$, and $Q_k := (P_k, g_{k-1}), k \in \mathbb{N}$. In the following, we present the sufficient descent property of $Q_k$ along the Lyapunov function $F$.

**Lemma.** Suppose that $\widehat{\mathcal{L}}_n$ is continuously differentiable and $\nabla\widehat{\mathcal{L}}_n$ is Lipschitz continuous with a constant $Lip > 0$. Let $\{Q_k\}$ be a sequence generated by SLBI with a finite initialization. If $0 < \alpha < \frac{2}{\kappa(Lip+\nu^{-1})}$, then

$$
F(Q_{k+1}) \leq F(Q_k) - \rho\|Q_{k+1} - Q_k\|_2^2,
$$

where $\rho := \frac{1}{\kappa} - \frac{\alpha(Lip+\nu^{-1})}{2}$.

*Proof.* By the optimality condition of (15a) and also the inclusion $p_k = [0, g_k]^T \in \partial R(P_k)$, there holds

$$
\kappa(\alpha\nabla\bar{\mathcal{L}}(P_k) + p_{k+1} - p_k) + P_{k+1} - P_k = 0,
$$

which implies

$$
-\langle\alpha\nabla\bar{\mathcal{L}}(P_k), P_{k+1} - P_k\rangle = \kappa^{-1}\|P_{k+1} - P_k\|_2^2 + D(\Gamma_{k+1}, \Gamma_k)
\tag{20}
$$

where

$$
D(\Gamma_{k+1}, \Gamma_k) := \langle g_{k+1} - g_k, \Gamma_{k+1} - \Gamma_k\rangle.
$$

Noting that $\bar{\mathcal{L}}(P) = \widehat{\mathcal{L}}_n(W) + \frac{1}{2\nu}\|W - \Gamma\|_2^2$ and by the Lipschitz continuity of $\nabla\widehat{\mathcal{L}}_n(W)$ with a constant $Lip > 0$ implies $\nabla\bar{\mathcal{L}}$ is Lipschitz continuous with a constant $Lip + \nu^{-1}$. This implies

$$
\bar{\mathcal{L}}(P_{k+1}) \leq \bar{\mathcal{L}}(P_k) + \langle\nabla\bar{\mathcal{L}}(P_k), P_{k+1} - P_k\rangle + \frac{Lip + \nu^{-1}}{2}\|P_{k+1} - P_k\|_2^2.
$$

Substituting the above inequality into (20) yields

$$
\alpha\bar{\mathcal{L}}(P_{k+1}) + D(\Gamma_{k+1}, \Gamma_k) + \rho\|P_{k+1} - P_k\|_2^2 \leq \alpha\bar{\mathcal{L}}(P_k).
\tag{21}
$$

Adding some terms in both sides of the above inequality and after some reformulations implies

$$
\alpha\bar{\mathcal{L}}(P_{k+1}) + B_\Omega^{g_k}(\Gamma_{k+1}, \Gamma_k)
\tag{22}
$$
$$
\leq \alpha\bar{\mathcal{L}}(P_k) + B_\Omega^{g_{k-1}}(\Gamma_k, \Gamma_{k-1}) - \rho\|P_{k+1} - P_k\|_2^2 - \left(D(\Gamma_{k+1}, \Gamma_k) + B_\Omega^{g_{k-1}}(\Gamma_k, \Gamma_{k-1}) - B_\Omega^{g_k}(\Gamma_{k+1}, \Gamma_k)\right)
$$
$$
= \alpha\bar{\mathcal{L}}(P_k) + B_\Omega^{g_{k-1}}(\Gamma_k, \Gamma_{k-1}) - \rho\|P_{k+1} - P_k\|_2^2 - B_\Omega^{g_{k+1}}(\Gamma_k, \Gamma_{k-1}) - B_\Omega^{g_{k-1}}(\Gamma_k, \Gamma_{k-1}),
$$

where the final equality holds for $D(\Gamma_{k+1}, \Gamma_k) - B_\Omega^{g_k}(\Gamma_{k+1}, \Gamma_k) = B_\Omega^{g_{k+1}}(\Gamma_k, \Gamma_{k-1})$. That is,

$$F(Q_{k+1}) \leq F(Q_k) - \rho\|P_{k+1} - P_k\|_2^2 - B_\Omega^{g_{k+1}}(\Gamma_k, \Gamma_{k-1}) - B_\Omega^{g_{k-1}}(\Gamma_k, \Gamma_{k-1}) \qquad (23)$$
$$\leq F(Q_k) - \rho\|P_{k+1} - P_k\|_2^2, \qquad (24)$$

where the final inequality holds for $B_\Omega^{g_{k+1}}(\Gamma_k, \Gamma_{k-1}) \geq 0$ and $B_\Omega^{g_{k-1}}(\Gamma_k, \Gamma_{k-1}) \geq 0$. Thus, we finish the proof of this lemma. $\square$

Based on Lemma A.4, we directly obtain the following lemma.

**Lemma 1.** *Suppose that assumptions of Lemma A.4 hold. Suppose further that Assumption 1 (b)-(d) hold. Then*

(i) *both $\alpha\{\bar{\mathcal{L}}(P_k)\}$ and $\{F(Q_k)\}$ converge to the same finite value, and $\lim_{k\to\infty} B_\Omega^{g_k}(\Gamma_{k+1}, \Gamma_k) = 0$.*

(ii) *the sequence $\{(W_k, \Gamma_k, g_k)\}$ is bounded,*

(iii) *$\lim_{k\to\infty} \|P_{k+1} - P_k\|_2^2 = 0$ and $\lim_{k\to\infty} D(\Gamma_{k+1}, \Gamma_k) = 0$,*

(iv) *$\frac{1}{K}\sum_{k=0}^{K} \|P_{k+1} - P_k\|_2^2 \to 0$ at a rate of $\mathcal{O}(1/K)$.*

*Proof.* By (21), $\bar{\mathcal{L}}(P_k)$ is monotonically decreasing due to $D(\Gamma_{k+1}, \Gamma_k) \geq 0$. Similarly, by (24), $F(Q^k)$ is also monotonically decreasing. By the lower boundedness assumption of $\hat{\mathcal{L}}_n(W)$, both $\bar{\mathcal{L}}(P)$ and $F(Q)$ are lower bounded by their definitions, i.e., (2) and (9), respectively. Therefore, both $\{\bar{\mathcal{L}}(P_k)\}$ and $\{F(Q_k)\}$ converge, and it is obvious that $\lim_{k\to\infty} F(Q_k) \geq \lim_{k\to\infty} \alpha\bar{\mathcal{L}}(P_k)$. By (23),

$$B_\Omega^{g_{k-1}}(\Gamma_k, \Gamma_{k-1}) \leq F(Q_k) - F(Q_{k+1}), \ k = 1, \ldots.$$

By the convergence of $F(Q_k)$ and the nonnegativeness of $B_\Omega^{g_{k-1}}(\Gamma_k, \Gamma_{k-1})$, there holds

$$\lim_{k\to\infty} B_\Omega^{g_{k-1}}(\Gamma_k, \Gamma_{k-1}) = 0.$$

By the definition of $F(Q_k) = \alpha\bar{\mathcal{L}}(P_k) + B_\Omega^{g_{k-1}}(\Gamma_k, \Gamma_{k-1})$ and the above equality, it yields

$$\lim_{k\to\infty} F(Q_k) = \lim_{k\to\infty} \alpha\bar{\mathcal{L}}(P_k).$$

Since $\hat{\mathcal{L}}_n(W)$ has bounded level sets, then $W_k$ is bounded. By the definition of $\bar{\mathcal{L}}(W, \Gamma)$ and the finiteness of $\bar{\mathcal{L}}(W_k, \Gamma_k)$, $\Gamma_k$ is also bounded due to $W_k$ is bounded. The boundedness of $g_k$ is due to $g_k \in \partial\Omega(\Gamma_k)$, condition (d), and the boundedness of $\Gamma_k$.

By (24), summing up (24) over $k = 0, 1, \ldots, K$ yields

$$\sum_{k=0}^{K} \left( \rho\|P_{k+1} - P_k\|^2 + D(\Gamma_{k+1}, \Gamma_k) \right) < \alpha\bar{\mathcal{L}}(P_0) < \infty. \qquad (25)$$

Letting $K \to \infty$ and noting that both $\|P_{k+1} - P_k\|^2$ and $D(\Gamma_{k+1}, \Gamma_k)$ are nonnegative, thus

$$\lim_{k\to\infty} \|P_{k+1} - P_k\|^2 = 0, \quad \lim_{k\to\infty} D(\Gamma_{k+1}, \Gamma_k) = 0.$$

Again by (25),

$$\frac{1}{K}\sum_{k=0}^{K} \left( \rho\|P_{k+1} - P_k\|^2 + D(\Gamma_{k+1}, \Gamma_k) \right) < K^{-1}\alpha\bar{\mathcal{L}}(P_0),$$

which implies $\frac{1}{K}\sum_{k=0}^{K} \|P_{k+1} - P_k\|^2 \to 0$ at a rate of $\mathcal{O}(1/K)$. $\square$

## A.5 RELATIVE ERROR PROPERTY

In this subsection, we provide the bound of subgradient by the discrepancy of two successive iterates. By the definition of $F$ (9),

$$H_{k+1} := \begin{pmatrix} \alpha\nabla_W\bar{\mathcal{L}}(W_{k+1},\Gamma_{k+1}) \\ \alpha\nabla_\Gamma\bar{\mathcal{L}}(W_{k+1},\Gamma_{k+1}) + g_{k+1} - g_k \\ \Gamma_k - \Gamma_{k+1} \end{pmatrix} \in \partial F(Q_{k+1}), \ k \in \mathbb{N}. \tag{26}$$

**Lemma.** Under assumptions of Lemma 1, then

$$\|H_{k+1}\| \le \rho_1\|Q_{k+1} - Q_k\|, \text{ for } H_{k+1} \in \partial F(Q_{k+1}), \ k \in \mathbb{N},$$

where $\rho_1 := 2\kappa^{-1} + 1 + \alpha(Lip + 2\nu^{-1})$. Moreover, $\frac{1}{K}\sum_{k=1}^K \|H_k\|^2 \to 0$ at a rate of $\mathcal{O}(1/K)$.

*Proof.* Note that

$$\nabla_W\bar{\mathcal{L}}(W_{k+1},\Gamma_{k+1}) = (\nabla_W\bar{\mathcal{L}}(W_{k+1},\Gamma_{k+1}) - \nabla_W\bar{\mathcal{L}}(W_{k+1},\Gamma_k)) \tag{27}$$
$$+ (\nabla_W\bar{\mathcal{L}}(W_{k+1},\Gamma_k) - \nabla_W\bar{\mathcal{L}}(W_k,\Gamma_k)) + \nabla_W\bar{\mathcal{L}}(W_k,\Gamma_k).$$

By the definition of $\bar{\mathcal{L}}$ (see (2)),

$$\|\nabla_W\bar{\mathcal{L}}(W_{k+1},\Gamma_{k+1}) - \nabla_W\bar{\mathcal{L}}(W_{k+1},\Gamma_k)\| = \nu^{-1}\|\Gamma_k - \Gamma_{k+1}\|,$$
$$\|\nabla_W\bar{\mathcal{L}}(W_{k+1},\Gamma_k) - \nabla_W\bar{\mathcal{L}}(W_k,\Gamma_k)\| = \|(\nabla\widehat{\mathcal{L}}_n(W_{k+1}) - \nabla\widehat{\mathcal{L}}_n(W_k)) + \nu^{-1}(W_{k+1} - W_k)\|$$
$$\le (Lip + \nu^{-1})\|W_{k+1} - W_k\|,$$

where the last inequality holds for the Lipschitz continuity of $\nabla\widehat{\mathcal{L}}_n$ with a constant $Lip > 0$, and by (16a),

$$\|\nabla_W\bar{\mathcal{L}}(W_k,\Gamma_k)\| = (\kappa\alpha)^{-1}\|W_{k+1} - W_k\|.$$

Substituting the above (in)equalities into (27) yields

$$\|\nabla_W\bar{\mathcal{L}}(W_{k+1},\Gamma_{k+1})\| \le [(\kappa\alpha)^{-1} + Lip + \nu^{-1}] \cdot \|W_{k+1} - W_k\| + \nu^{-1}\|\Gamma_{k+1} - \Gamma_k\|$$

Thus,

$$\|\alpha\nabla_W\bar{\mathcal{L}}(W_{k+1},\Gamma_{k+1})\| \le [\kappa^{-1} + \alpha(Lip + \nu^{-1})] \cdot \|W_{k+1} - W_k\| + \alpha\nu^{-1}\|\Gamma_{k+1} - \Gamma_k\|. \tag{28}$$

By (16c), it yields

$$g_{k+1} - g_k = \kappa^{-1}(\Gamma_k - \Gamma_{k+1}) - \alpha\nabla_\Gamma\bar{\mathcal{L}}(W_k,\Gamma_k).$$

Noting that $\nabla_\Gamma\bar{\mathcal{L}}(W_k,\Gamma_k) = \nu^{-1}(\Gamma_k - W_k)$, and after some simplifications yields

$$\|\alpha\nabla_\Gamma\bar{\mathcal{L}}(W_{k+1},\Gamma_{k+1}) + g_{k+1} - g_k\| = \|(\kappa^{-1} - \alpha\nu^{-1}) \cdot (\Gamma_k - \Gamma_{k+1}) + \alpha\nu^{-1}(W_k - W_{k+1})\|$$
$$\le \alpha\nu^{-1}\|W_k - W_{k+1}\| + (\kappa^{-1} - \alpha\nu^{-1})\|\Gamma_k - \Gamma_{k+1}\|, \tag{29}$$

where the last inequality holds for the triangle inequality and $\kappa^{-1} > \alpha\nu^{-1}$ by the assumption.

By (28), (29), and the definition of $H_{k+1}$ (26), there holds

$$\|H_{k+1}\| \le [\kappa^{-1} + \alpha(Lip + 2\nu^{-1})] \cdot \|W_{k+1} - W_k\| + (\kappa^{-1} + 1)\|\Gamma_{k+1} - \Gamma_k\|$$
$$\le [2\kappa^{-1} + 1 + \alpha(Lip + 2\nu^{-1})] \cdot \|P_{k+1} - P_k\| \tag{30}$$
$$\le [2\kappa^{-1} + 1 + \alpha(Lip + 2\nu^{-1})] \cdot \|Q_{k+1} - Q_k\|.$$

By (30) and Lemma 1(iv), $\frac{1}{K}\sum_{k=1}^K \|H_k\|^2 \to 0$ at a rate of $\mathcal{O}(1/K)$.

This finishes the proof of this lemma. $\square$

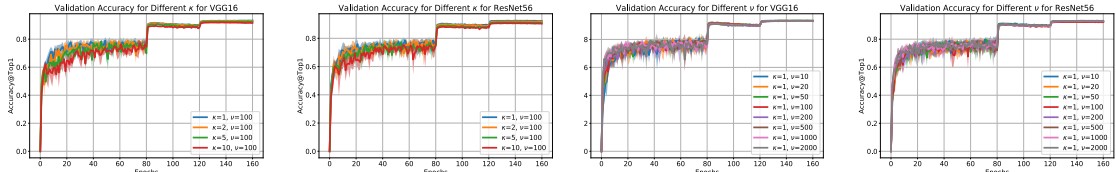

Figure 6: Validation curves of dense models $W_t$ for different $\kappa$ and $\nu$. For SLBI we find that the model accuracy is robust to the hyperparameters both in terms of convergence rate and generalization ability. Here validation accuracy means the accuracy on test set of Cifar10. The first one is the result for VGG16 ablation study on $\kappa$, the second one is the result for ResNet56 ablation study on $\kappa$, the third one is the result for VGG16 ablation study on $\nu$ and the forth one is the result for ResNet56 ablation study on $\nu$.

| Type | Model | $\kappa = 1$ | $\kappa = 2$ | $\kappa = 5$ | $\kappa = 10$ | SGD |
|------|-------|--------------|--------------|--------------|---------------|------|
| Full | Vgg16 | 93.46 | 93.27 | 92.77 | 92.03 | 93.57 |
| | ResNet56 | 92.71 | 92.18 | 91.50 | 90.92 | 93.08 |
| Sparse | Vgg16 | 93.31 | 93.00 | 92.36 | 76.25 | - |
| | ResNet56 | 92.37 | 91.85 | 89.48 | 87.02 | - |

Table 2: This table shows results for different $\kappa$, the results are all the best test accuracy. Here we test two widely-used models: VGG16 and ResNet56 on Cifar10. For results in this table, we keep $\nu = 100$. Full means that we use the trained model weights directly, Sparse means the model weights are combined with mask generated by $\Gamma$ support. Sparse result has no finetuning process, the result is comparable to its Full counterpart. For this experiment, we propose that $\kappa = 1$ is a good choice. For all the model, we train for 160 epochs with initial learning rate (lr) of 0. 1 and decrease by 0.1 at epoch 80 and 120.

# B  SUPPLEMENTARY EXPERIMENTS

## B.1  ABLATION STUDY OF VGG16 AND RESNET56 ON CIFAR10

To further study the influence of hyperparameters, we record performance of $W_t$ for each epoch $t$ with different combinations of hyperparameters. The experiments is conducted 5 times each, we show the mean in the table, the standard error can be found in the corresponding figure. We perform experiments on Cifar10 and two commonly used network VGG16 and ResNet56.

On $\kappa$, we keep $\nu = 100$ and try $\kappa = 1, 2, 5, 10$, the validation curves of models $W_t$ are shown in Fig. 6 and Table 2 summarizes the mean accuracies. Table 3 summarizes best validation accuracies achieved at some epochs, together with their sparsity rates. These results show that larger kappa leads to slightly lower validation accuracies, where the numerical results are shown in Table 2 . We can find that $\kappa = 1$ achieves the best test accuracy.

On $\nu$, we keep $\kappa = 1$ and try $\nu = 10, 20, 50, 100, 200, 500, 1000, 2000$ the validation curve and mean accuracies are show in Fig. 6 and Table 4. Table 5 summarizes best validation accuracies achieved at some epochs, together with their sparsity rates. By carefully tuning $\nu$ we can achieve similar or even better results compared to SGD. Different from $\kappa$, $\nu$ has less effect on the generalization performance. By tuning it carefully, we can even get a sparse model with slightly better performance than SGD trained model.

# C  COMPUTATIONAL COST OF SPLITLBI

We further compare the computational cost of different optimizers: SGD (Mom), SplitLBI (Mom) and Adam (Naive). We test each optimizer on one GPU, and all the experiments are done on one GTX2080. For computational cost, we judge them from two aspects : GPU memory usage and time needed for one batch. The batch size here is 64, experiment is performed on VGG-16 as shown in Table 6.

| Model | | Ep20 | | Ep40 | | Ep80 | | Ep160 | |
|---|---|---|---|---|---|---|---|---|---|
| | Term | Sparsity | Acc | Spasity | Acc | Spasity | Acc | Spasity | Acc |
| Vgg16 | $\kappa = 1$ | 96.62 | 71.51 | 96.62 | 76.92 | 96.63 | 77.48 | 96.63 | 93.31 |
| | $\kappa = 2$ | 51.86 | 72.98 | 71.99 | 73.64 | 75.69 | 74.54 | 75.72 | 93.00 |
| | $\kappa = 5$ | 8.19 | 10.00 | 17.64 | 34.25 | 29.76 | 69.92 | 30.03 | 92.36 |
| | $\kappa = 10$ | 0.85 | 10.00 | 6.62 | 10.00 | 12.95 | 38.38 | 13.26 | 76.25 |
| | Term | Sparsity | Acc | Spasity | Acc | Spasity | Acc | Spasity | Acc |
| ResNet56 | $\kappa = 1$ | 96.79 | 73.50 | 96.87 | 75.27 | 96.69 | 77.47 | 99.68 | 92.37 |
| | $\kappa = 2$ | 76.21 | 72.85 | 81.41 | 74.72 | 84.17 | 75.64 | 84.30 | 91.85 |
| | $\kappa = 5$ | 36.58 | 60.43 | 53.07 | 76.00 | 57.48 | 75.67 | 57.74 | 89.48 |
| | $\kappa = 10$ | 3.12 | 10.20 | 29.43 | 53.36 | 41.18 | 74.56 | 41.14 | 87.02 |

Table 3: Sparsity rate and validation accuracy for different $\kappa$ at different epochs. Here we pick the test accuracy for specific epoch. In this experiment, we keep $\nu = 100$. We pick epoch 20, 40, 80 and 160 to show the growth of sparsity and sparse model accuracy. Here Sparsity is defined in Sec. 4, and Acc means the test accuracy for sparse model. A sparse model is a model at designated epoch $t$ combined with the mask as the support of $\Gamma_t$.

| Type | Model | $\nu = 10$ | $\nu = 20$ | $\nu = 50$ | $\nu = 100$ | $\nu = 200$ | $\nu = 500$ | $\nu = 1000$ | $\nu = 2000$ | SGD |
|---|---|---|---|---|---|---|---|---|---|---|
| Full | Vgg16 | 93.66 | 93.59 | 93.57 | 93.39 | 93.38 | 93.35 | 93.43 | 93.46 | 93.57 |
| | ResNet56 | 93.12 | 92.68 | 92.78 | 92.45 | 92.95 | 93.11 | 93.16 | 93.31 | 93.08 |
| Sparse | Vgg16 | 93.39 | 93.42 | 93.39 | 93.23 | 93.21 | 93.01 | 92.68 | 10 | - |
| | ResNet56 | 92.81 | 92.19 | 92.40 | 92.10 | 92.68 | 92.81 | 92.84 | 88.96 | - |

Table 4: Results for different $\nu$, the results are all the best test accuracy. Here we test two widely-used model : VGG16 and ResNet56 on Cifar10. For results in this table, we keep $\kappa = 1$. Full means that we use the trained model weights directly, Sparse means the model weights are combined with mask generated by $\Gamma$ support. Sparse result has no finetuning process, the result is comparable to its Full counterpart. For all the model, we train for 160 epochs with initial learning rate (lr) of 0.1 and decrease by 0.1 at epoch 80 and 120.

| Model | | Ep20 | | Ep40 | | Ep80 | | Ep160 | |
|---|---|---|---|---|---|---|---|---|---|
| | Term | Sparsity | Acc | Spasity | Acc | Spasity | Acc | Spasity | Acc |
| Vgg16 | $\nu = 10$ | 96.64 | 71.07 | 96.64 | 77.70 | 96.65 | 79.46 | 96.65 | 93.34 |
| | $\nu = 20$ | 96.64 | 69.11 | 96.64 | 77.63 | 96.65 | 77.08 | 96.65 | 93.42 |
| | $\nu = 50$ | 96.64 | 74.91 | 96.65 | 74.21 | 96.65 | 79.15 | 96.65 | 93.38 |
| | $\nu = 100$ | 96.64 | 74.82 | 96.64 | 73.22 | 96.64 | 78.09 | 96.64 | 93.23 |
| | $\nu = 200$ | 91.69 | 73.67 | 94.06 | 74.67 | 94.15 | 75.20 | 94.15 | 93.21 |
| | $\nu = 500$ | 18.20 | 10.00 | 59.94 | 67.88 | 82.03 | 78.69 | 82.32 | 93.01 |
| | $\nu = 1000$ | 6.43 | 10.00 | 17.88 | 10.00 | 49.75 | 61.31 | 51.21 | 92.68 |
| | $\nu = 2000$ | 0.22 | 10.00 | 6.89 | 10.00 | 18.15 | 10.00 | 19.00 | 10.00 |
| | Term | Sparsity | Acc | Spasity | Acc | Spasity | Acc | Spasity | Acc |
| ResNet56 | $\nu = 10$ | 99.97 | 73.37 | 99.95 | 71.64 | 99.74 | 76.46 | 99.74 | 92.81 |
| | $\nu = 20$ | 99.97 | 72.58 | 99.84 | 74.16 | 99.69 | 72.37 | 99.72 | 92.19 |
| | $\nu = 50$ | 99.96 | 70.72 | 99.89 | 73.96 | 99.79 | 74.93 | 99.77 | 92.40 |
| | $\nu = 100$ | 96.31 | 73.63 | 96.63 | 75.79 | 96.55 | 72.94 | 96.57 | 92.10 |
| | $\nu = 200$ | 91.98 | 75.30 | 94.38 | 72.13 | 94.87 | 73.75 | 94.88 | 92.68 |
| | $\nu = 500$ | 74.44 | 65.58 | 90.00 | 74.12 | 92.96 | 71.91 | 92.99 | 92.81 |
| | $\nu = 1000$ | 24.32 | 10.85 | 75.68 | 70.23 | 88.56 | 79.67 | 88.80 | 92.48 |
| | $\nu = 2000$ | 0.65 | 10.00 | 26.66 | 13.30 | 74.98 | 70.38 | 75.92 | 88.95 |

Table 5: Sparsity rate and validation accuracy for different $\nu$ at different epochs. Here we pick the test accuracy for specific epoch. In this experiment, we keep $\kappa = 1$. We pick epoch 20, 40, 80 and 160 to show the growth of sparsity and sparse model accuracy. Here Sparsity is defined in Sec. 4 as the percentage of nonzero parameters, and Acc means the test accuracy for sparse model. A sparse model is a model at designated epoch $t$ combined with mask as the support of $\Gamma_t$.

| optimizer | SGD | SLBI | Adam |
|---|---|---|---|
| Mean Batch Time | 0.0197 | 0.0221 | 0.0210 |
| GPU Memory | 1161MB | 1459MB | 1267MB |

Table 6: Computational and Memory Costs. ( And GPU memory means the whole memory model weights and the activation cache. )

| Layer | FC1 | FC2 | FC3 |
|---|---|---|---|
| Sparsity | 0.049 | 0.087 | 0.398 |
| Number of Weights | 235200 | 30000 | 1000 |

Table 7: This table shows the sparsity for every layer of Lenet-3. Here sparsity is defined in Sec. 4, number of weights denotes the total number of parameters in the designated layer. It is interesting that the $\Gamma$ tends to put lower sparsity on layer with more parameters.

## D    FINE-TUNING OF SPARSE SUBNETWORKS

We design the experiment on MNIST, inspired by Frankle & Carbin (2019). Here, we explore the subnet obtained by $\Gamma_T$ after $T = 100$ epochs of training. As in Frankle et al. (2019), we adopt the "rewind" trick: re-loading the subnet mask of $\Gamma_{100}$ at different epochs, followed by fine-tuning. In particular, along the training paths, we reload the subnet models at Epoch 0, Epoch 30, 60, 90, and 100, and further fine-tune these models by SplitLBI (Mom-Wd). All the models use the same initialization and hence the subnet model at Epoch 0 gives the retraining with the same random initialization as proposed to find winning tickets of lottery in Frankle & Carbin (2019). We will denote the rewinded fine-tuned model at epoch 0 as (Lottery), and those at epoch 30, 60, 90, and 100, as F-epoch30, F-epoch60, F-epoch90, and F-epoch100, respectively. Three networks are studied here – LeNet-3, Conv-2, and Conv-4. LeNet-3 removes one convolutional layer of LeNet-5; and it is thus less over-parameterized than the other two networks. Conv-2 and Conv-4, as the scaled-down variants of VGG family as done in Frankle & Carbin (2019), have two and four fully-connected layers, respectively, followed by max-pooling after every two convolutional layer.

The whole sparsity for Lenet-3 is 0.055, Conv-2 is 0.0185, and Conv-4 is 0.1378. Detailed sparsity for every layer of the model is shown in Table 7, 8, 9. We find that fc-layers are sparser than conv-layers.

We compare SplitLBI variants to the SGD (Mom-Wd) and SGD (Lottery) (Frankle & Carbin, 2019) in the same structural sparsity and the results are shown in Fig. 7. In this exploratory experiment, one can see that for overparameterized networks – Conv-2 and Conv-4, fine-tuned rewinding subnets – F-epoch30, F-epoch60, F-epoch90, and F-epoch100, can produce *better* results than the full models; while for the less over-parameterized model LeNet-3, fine-tuned subnets may achieve less yet still comparable performance to the dense models and remarkably better than the retrained sparse subnets from beginning (i.e. SplitLBI/SGD (Lottery)). These phenomena suggest that the subnet architecture disclosed by structural sparsity parameter $\Gamma_T$ is valuable, for fine-tuning sparse models with comparable or even better performance than the dense models of $W_T$.

## E    RETRAINING OF SPARSE SUBNETS FOUND BY SPLITLBI (LOTTERY)

Here we provide more details on the experiments in Fig. 5. Table 10 gives the details on hyper-parameter setting. Moreover, Figure 8 provides the sparsity variations during SplitLBI training in Fig. 5.

| Layer | Conv1 | Conv2 | FC1 | FC2 | FC3 |
|---|---|---|---|---|---|
| Sparsity | 0.9375 | 1 | 0.0067 | 0.0284 | 0.1551 |
| Number of Weights | 576 | 36864 | 3211264 | 65536 | 2560 |

Table 8: This table shows the sparsity for every layer of Conv-2. Here sparsity is defined in Sec. 4, number of weights denotes the total number of parameters in the designated layer. The sparsity is more significant in fully connected (FC) layers than convolutional layers.

| Layer | Conv1 | Conv2 | Conv3 | Conv4 | FC1 | FC2 | FC3 |
|---|---|---|---|---|---|---|---|
| Sparsity | 0.921875 | 1 | 1 | 1 | 0.0040 | 0.0094 | 0.1004 |
| Number of Weights | 576 | 36864 | 73728 | 147456 | 1605632 | 65536 | 2560 |

Table 9: This table shows the sparsity for every layer of Conv-4. Here sparsity is defined in Sec. 4, number of weights denotes the total number of parameters in the designated layer. Most of the convolutional layers are kept while the FC layers are very sparse.

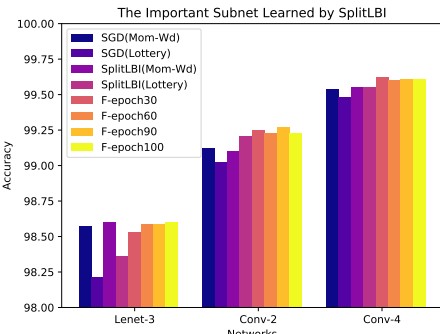

Figure 7: Fine-tuning of sparse subnets learned by SplitLBI may achieve comparable or better performance than dense models. F-epoch$k$ indicates the fine-tuned model comes from the Epoch $k$. SplitLBI (Lottery) and SGD (Lottery) use the same sparsity rate for each layer and the same initialization for retrain.

| Network | Penalty | Optimizer | $\alpha$ | $\nu$ | $\kappa$ | $\lambda$ | Momentum | Nesterov |
|---|---|---|---|---|---|---|---|---|
| VGG-16 | Group Lasso | SLBI | 0.1 | 100 | 1 | 0.1 | 0.9 | Yes |
| ResNet-56 | Group Lasso | SLBI | 0.1 | 100 | 1 | 0.05 | 0.9 | Yes |
| VGG-16(Lasso) | Lasso | SLBI | 0.1 | 500 | 1 | 0.05 | 0.9 | Yes |
| ResNet-50(Lasso) | Lasso | SLBI | 0.1 | 200 | 1 | 0.03 | 0.9 | Yes |

Table 10: Hyperparameter setting for the experiments in Figure 5.

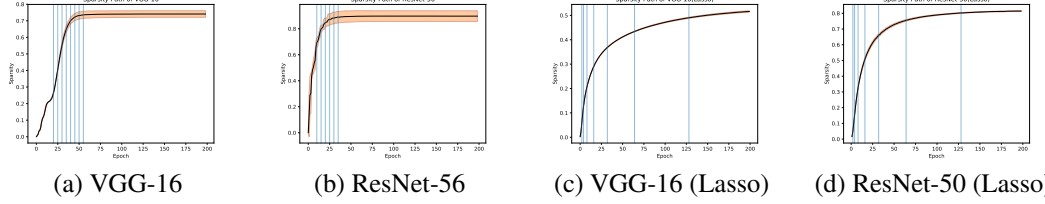

| (a) VGG-16 | (b) ResNet-56 | (c) VGG-16 (Lasso) | (d) ResNet-50 (Lasso) |

Figure 8: Sparsity changing during training process of SplitLBI (Lottery) for VGG and ResNets (corresponding to Fig. 5). We calculate the sparsity in every epoch and repeat five times. The black curve represents the mean of the sparsity and shaded area shows the standard deviation of sparsity. The vertical blue line shows the epochs that we choose to early stop. We choose the log-scale epochs for achieve larger range of sparsity.

