# OpenReview forum: "Split LBI for Deep Learning: Structural Sparsity via Differential Inclusion Paths"
_ICLR.cc/2020/Conference — Reject_

### Official Review · AnonReviewer3 · 2019-10-23
**Official Blind Review #3**

**Rating:** 8

**Review:**

This paper studies the problem of finding important structural sparsity of over-parameterized neural networks. Compared to the traditional pruning and distillation based compression methods, the author propose a novel optimization based algorithm called SplitLBI, which can efficiently and effectively find the important subnetwork architecture. In addition, the authors provide the convergence guarantee of the proposed method for solving nonconvex optimization problems under certain conditions. Thorough experiments and ablation studies validate the advantage of the proposed method.

The contributions of this paper are as follows:
1. A novel optimization based method for finding important sparse structure of large-scale neural networks using the idea of coupling the learning of weight matrix and sparsity constraints.
2. The convergence guarantee of the proposed method for solving nonconvex optimization is established based on a novel Lyapunov function.
3.Extensive experiments show that the proposed algorithm can: (a) train over-parameterized neural networks to achieve the state-of-the-art performances on image classification tasks; (b) find sparse networks with competitive networks; and (c) early stopping plus retrain from scratch can achieve similar or better  performance than existing compression method.

I believe that this work is very interesting and will be of interest to the ICLR community. The presentation of this paper is clear and the idea of coupling the gradient descent and mirror descent is very interesting. In addition, the empirical evaluations, including the sensitivity of the parameters and computation and memory costs, of the proposed method are solid.

Minor comments:
1.There is no definition of KL function in the main content.
2.Before presenting Corollary 1, please specify the g_k as well as the updates for training neural networks.
3.Is it possible to extend the convergence guarantee to the ReLU activation function since the architectures considered in experiments using such activation.
4.In experiments it is better to present the convergence of the training loss to validate the convergence results.
5.In Figure 2, what is SLBI-10 and SLBI-1?

Update: Thanks for your clarification, and I recommend acceptance of this work.

**Experience Assessment:**

I have read many papers in this area.

**Review Assessment: Checking Correctness Of Derivations And Theory:**

I assessed the sensibility of the derivations and theory.

**Review Assessment: Checking Correctness Of Experiments:**

I carefully checked the experiments.

**Review Assessment: Thoroughness In Paper Reading:**

I read the paper at least twice and used my best judgement in assessing the paper.

---

> ### Author Response · Authors · 2019-11-15
> **For Reviewer #3**
>
> Q1.There is no definition of KL function in the main content.
>
> A. Thanks. Yes, due to page limit, we put the of Kurdyka-Lojasiewicz (KL) property and KL function in the Appendix (A1 and A2) respectively. We add the reference of Kurdyka-Lojasiewicz property, and referring to A1, and A2 of our paper
>
>
> Q2.Before presenting Corollary 1, please specify the g_k as well as the updates for training neural networks.
>
> A.The g_k is the subgradient of $\Omega(\Gamma)$ and updates as follows:
> g_{k+1}=g_{k}-\kappa^{-1}(\Gamma_{k+1}-\Gamma_{k}+\kappa\alpha\cdot\nabla_{\Gamma}\bar{\mathcal{L}}(W_{k},\Gamma_{k})). Together with W_k and \Gamma_k, the update rule of (W_k,\Gamma_k,g_k) is the equivalent form with Splitlbi, Eq. (1). We introduce it for the convenience of proof, the detail of which is left in Appendix A.1.
>
>
> Q3.Is it possible to extend the convergence guarantee to the ReLU activation function since the architectures considered in experiments using such activation.
>
> A. Currently, our proof relies on the condition that the loss function, i.e., the activation function is smooth, which applies to softplus function (i.e., 1/c log(1 + e^{ct}) ) that can approximate to relu with large value of c.
>
> Q4.In experiments it is better to present the convergence of the training loss to validate the convergence results.
>
> A. Thanks for your suggestions. Training loss curves are provided in Figure 3 in revision, where the curves can be seen going flat, indicating function value convergence.
>
>
> Q5.In Figure 2, what is SLBI-10 and SLBI-1?
>
> A.The SLBI-10 and SLBI-1 refers to SplitLBI with kappa = 10 and kappa = 1, respectively. Particularly, we add the sentence “ The ”SLBI-10” (”SLBI-1”) in the right figure refers to SplitLBI with \kappa = 10 and  \kappa = 1, respectively in Figure 2.

---

### Official Review · AnonReviewer2 · 2019-10-25
**Official Blind Review #2**

**Rating:** 6

**Review:**


Edit after rebuttals: I have read all other reviews and rebuttals and maintain my assessment.
----

I believe this paper should have been desk-rejected, as it is 9p long in the main text (above the suggested limit of 8), plus 4p of bibliography, for a total of 13p, above the hard limit of 10p. It is concerning that all figures are so squeezed as to be unreadable in print version. In the remainder of my review, as well as my rating, I will ignore the length issue, and leave it to the AC to include this fact.

This paper is concerned with methods to enforce sparsity in neural networks (specially CNN for image classification) at training time. It is motivated by the issue that L1 regularisation violates incoherence and irrepresentability conditions (cf sec1 and sec2 below eq3). The approach taken to encourage structural sparsity is to formulate the training dynamics in terms not only of the neural network weights $W$, but also of their sparse version $V$.

I recommend (other than for the length issue) this paper for publication. My review needs to be relativised as I am not familiar with the literature on inverse scale spaces. Factors for my recommendation are: (1) the paper is clearly written (apart from an important language issue, cf below) and laid out, (2) the problem is well-motivated and contextualised (although I cannot judge novelty), (3) experiments, including analytic and ablation experiments, seem well designed and conducted, (4) experimental evidence shows that the proposed method is efficient, (5) the formal treatment is strong and thorough, with relevant literature cited.

Suggestions for improvement
- faulty language definitely stands in the way of understanding, in several places. There is about one syntax error in every paragraph of the paper. Sentences are too long as they span several lines, and should be broken up for clarity. In particular, the use of "that" and gerunds vs present tense is mostly erroneous. Eg: "while the other set of parameters learning", "structural sparsity set $\Gamma$ that", "comparable test accuracy as", "that the whole iterative sequence", "which drives $W_t$ closely following", "whose global convergence condition to be shown" etc.
- to allow non-specialists to read this paper, you should more clearly explain and illustrate the spaces in which different variables are, starting with $V$ and $\Gamma$.
- the paper lacks a conclusion which should bring together several strands of argumentation
- I strongly suggest publishing the source code, not just promising to release them upon request. The methods described here require some skill to be coded, especially I cannot see how to easily re-use existing library components; this makes it hard for the ideas in this paper to be applied and diffused.
- it is insufficient to state that "official source code" was used, as in table 1; this should definitely state the (Github or other) URL where the code is available

**Experience Assessment:**

I do not know much about this area.

**Review Assessment: Checking Correctness Of Derivations And Theory:**

I assessed the sensibility of the derivations and theory.

**Review Assessment: Checking Correctness Of Experiments:**

I assessed the sensibility of the experiments.

**Review Assessment: Thoroughness In Paper Reading:**

I read the paper at least twice and used my best judgement in assessing the paper.

---

> ### Author Response · Authors · 2019-11-15
> **For Reviewer #2**
>
> 1, As to the pages, we actually try to follow the submission instructions as https://iclr.cc/Conferences/2020/CallForPapers, which specified that (1)The recommended paper length is 8 pages, with unlimited additional pages for citations. (2)There will be a strict upper limit of 10 pages for the main text. Reviewers will be instructed to apply a higher standard to papers in excess of 8 pages.  (3) Authors may use as many pages of appendices (after the bibliography) as they wish, but reviewers are not required to read these. In original submission we have 9 pages and the updated revision is within 10 pages, with all supporting material in appendices.
>
>
> 2, Thanks for your suggestions! We had revised our syntax errors accordingly in the new version.
>
> 3, We update and add explanations of these variables marked by red on page 3. Particularly, V is sub-gradient of $\bar{\Omega(\Gamma)}$ and $\Gamma$ is the augmented variable that models the structural sparsity of $\Gamma$. In mirror descent, a gradient descent of $V$ is understood as dual space descent, that is mapped to the primal space $\Gamma$ with structural sparsity.
>
>
> 4, We now add the conclusion to summarize our novelty as suggested. In particular, “In this paper, a parsimonious deep learning method is proposed based on differential inclusions of inverse scale spaces. Implemented by a variable splitting scheme, such a dynamics system can exploit over-parameterized models and structural sparsity simultaneously. Besides, its simple discretization, i.e., the SplitLBI, has a proven global convergence and hence can be employed to train deep networks. We have experimentally shown that it can achieve the state-of-the-art performance on many datasets including ImageNet-2012, with better interpretability than SGD. What's more, equipped with early stopping, such a structural sparsity can unveil the "winning tickets" -- the architecture of sub-network which after re-training can achieve comparable and even better accuracy than original dense network.”
>
> 5, The “official pytorch website” we mentioned in Table. 1 to report the accuracy of competitor models refers to https://pytorch.org/docs/stable/torchvision/models.html ; while the “official pytorch code” for realizing the competitor refers to https://github.com/pytorch/vision/tree/master/torchvision/models, based on which we develop the codes for our method.
>
> 6, We had cleaned up and add the anonymous github code link in the revised version. The codes will be formally released once accepted.

---

### Official Review · AnonReviewer4 · 2019-11-12
**Official Blind Review #4**

**Rating:** 3

**Review:**

Summary
=======
This paper aims to train sparse neural networks efficiently, by jointly optimizing the weights and sparsity structure of the network. It applies the Split Linear Bregman Iteration (Split LBI) method from [1] in a large-scale setting, to train deep neural networks.

The approach works by considering optimization in a joint space (W, \Gamma) consisting of the network weights W and a new set of parameters \Gamma that model the structural sparsity of the network.

The problem of learning sparse networks efficiently is important for modern applications that run on embedded devices, as well as for fast training on specialized hardware. I think the approach is interesting as a potential alternative to more expensive methods for finding sparse networks, such as NAS and the successive pruning & re-training approach of [2].

Overall, the paper pursues a promising direction to induce network sparsity, and presents some interesting results. However, there are several issues with the experiments and the structure/presentation of the paper that should be addressed.

Pros
====
* As far as I am aware, this is the first application of Split LBI to train deep neural networks.
* It shows that joint optimization of the weights and sparsity structure performs on par with baselines that only optimize weights, on MNIST, CIFAR-10, and ImageNet.
* It provides a global convergence analysis that shows that the weights optimized with Split LBI converge to a critical point of the training loss, regardless of initialization.
* It provides an ablation study for the two hyperparameters \kappa and \nu of Split LBI.
* I think the most interesting parts of the paper are those that examine the structural sparsity learned by Split LBI (Sections 4.3 and 4.4). In particular, the fact that Split LBI was able to match or outperform the test accuracy of several baselines (network slimming, soft filter pruning, and the method in Rethinking the Lottery Ticket Hypothesis) in a single training run (without re-training) is a nice result.

Issues
======
* The Split LBI method is presented as a novel contribution (in the abstract: "we propose a new approach based on differential inclusions of inverse scale spaces ..."), but it was already described in detail in a paper that they cite ([1]) published at NeurIPS 2016. The only theoretical contribution of this paper is section 3: Global convergence of Split LBI.

* The claim that "Split LBI demonstrates SOTA performance in large scale training on ImageNet" (from the abstract) is not correct, and needs to be qualified. This paper reports 70.55/89.56% (top 1 / top 5) accuracy; as far as I am aware, the current SOTA on ImageNet is [3], which achieves 86.4/98.0% (top 1 / top 5) accuracy. I think it would be better for this paper to argue that it achieves comparable performance to baselines with a particular architecture and training regime.

* The structure and presentation of the paper could be improved in several ways, outlined as follows:
    - Most of the Methodology section discusses the Split LBI method from prior work. I would encourage the authors to split the Methodology section into a separate Background section for Split LBI, followed by a new section specifically about applying Split-LBI to convolutional and fully-connected layers in neural networks.

    - The writing is missing some details and explanations that would be very helpful for readers. For example, it should clearly state that the dimension of \Gamma is the same as the dimension of the weights W.

    - It would also be good to expand the explanation about why SplitISS avoids the parameter correlation problem, and what it means for \Gamma to have an orthogonal design?

    - The figures are too small to be readable without a lot of zooming.

    - The paper ends abruptly, with no conclusion.

    - The appendix contains some useful material, but much of it is not referenced from the main paper. I think some parts of the appendix could be moved to the main paper, for example the comparison of computational and memory costs between Split LBI and SGD.

* I am not sure what the purpose of the comparisons between optimizers in Table 1 is. The motivation given in the abstract and introduction is to learn sparse networks online during optimization; it does not propose Split LBI as a new optimizer to compete with Adam. Couldn't one use the Adam update rule to optimize the weights in Split-LBI? I think it makes sense to compare Split LBI to standard training setups that do not enforce any sparsity, as well as to setups that use L1 and L2 regularization, but I do not think that Table 1 is set up correctly for this. Different optimizers are paired with different regularizers, and crucially the choice of hyperparameters is not discussed---how did you choose the coefficient of L1 regularization to be 1e-3? Additionally, many rows in Table 1 are missing data (e.g., the variants of Adam are only run for CIFAR-10).

* Regarding experiments, it is not clear which experiments the authors actually performed, for which they took the results from previously published papers, and sometimes where the results come from at all.

    - In Table 1, there is an asterix next to SGD-Mom-Wd that the authors say indicates "results from the official pytorch website." That would imply that the rest of the experiments were done by the authors (and the authors say in the caption "we use the official pytorch codes to run the competitors"). However, Table 2 (found in the Appendix, page 20), contains identical numbers and a sign # that, according to the authors, indicates results of their own experiments. That would mean that of all the SGD and Adam experiments shown in the table, the authors only performed SGD-naive and Adam-naive. Where do the other numbers come from? What does "official pytorch code to run the competitors" mean? Where is that code from?

    - Figure 4 contains baselines and SplitLBI results. Where do the numbers for the baselines come from? The caption mentions another paper, [5], but I did not find the source of the numbers in that paper. The caption seems to point to Table 9a in [5], but that table does not deal with Network Slimming, Soft-Filter Pruning, Scratch B, or Scratch-E. Additionally, Table 9a of [5] only contains results for VGG-16 and ResNet-50. Where do the baselines for ResNet-56 (Figure 4b) come from?

* How is the proximal objective in Eq. 5 optimized? That is, how do you compute the argmin?

* Figure 2 shows results for SLBI-1 and SLBI-10, but no discussion of what SLBI-1 and SLBI-10 mean. Also regarding Fig.2, the authors claim that "Filters learned by ImageNet prefer non-semantic texture rather than shape and color." How did the authors come to this conclusion? I looked carefully at the filter visualizations, and I cannot see a clear difference between the filters learned by Split LBI and SGD.

* The computation time comparison in Table 11 (Appendix E) is a bit strange, because it shows that Adam takes 2x as long as SGD, which does not align with my experience; in practice, the wall-clock time is nearly identical between Adam and SGD. It would be good to provide more details about how the time was measured. Also, does the memory comparison measure only the memory used for model parameters (W and \Gamma), or also activation memory? Shouldn't Split LBI use 2x the memory of SGD (if measuring only the weights)?

* In Figure 1, it looks like the initial magnitude of the filters is larger for SGD compared to Split LBI. Are the weights initialized in the same way? Also, why is the setup of the MNIST experiment in Fig.1 different from the setup in Table 1 (e.g., learning rate decay every 40 epochs vs every 30 epochs)? In addition, it looks like the first learning rate decay causes the filter weight magnitudes to flatten out and stay constant.

* What is the learning rate schedule used for the runs in Figure 3? It looks like the lr decays at epoch 80 and 120, but this is only mentioned in table captions in the appendix. This should be stated in the main paper. Also, why is this a different training setup from that used for Table 1?

I also noted that the authors do not intend to make the code public upon publication of the paper. On page 6, they state that "source codes will be released upon requests." At present, the preferred path is to make the code public upon publication of the paper.

Minor points
============
* In the caption of Figure 3, it says "The results are repeated for 5 times. Shaded area indicates the variance; and in each round, we keep the exactly same initialization for each model." What is different between the 5 runs if the initialization is the same?

* There are too many different colors used in Figures 1 and 2. Since the purple, green, and black boxes are important to see for figures 1 and 2, it is confusing to have to deal with additional blue, pink, and yellow boxes around every three.


[1] Huang et al., Split LBI: An iterative regularization path with structural sparsity. NeurIPS 2016.
[2] Frankle & Carbin, The Lottery Ticket Hypothesis: Finding sparse, trainable neural networks. ICLR 2019.
[3] Touvron et al., Fixing the train-test resolution discrepancy. https://arxiv.org/abs/1906.06423.
[4] He et al., Deep residual learning for image recognition. https://arxiv.org/abs/1512.03385.
[5] Liu et al., Rethinking the value of network pruning. ICLR 2019.



Post-rebuttal Update
====================

I thank the authors for their rebuttal, and for clarifying some details in the paper.

* I think the experiments on sparsity are interesting. More efficient ways to find good sparse networks are certainly of interest to the community.

* I appreciate that the authors released the source code.

* In summary, this paper applies Split-LBI to neural network training, and provides a global convergence result as one of the main contributions. Operationally, compared to the original Split LBI approach, it changes the loss function from squared error to cross entropy, and uses mini-batches for training, which are fairly straightforward.

* One important issue with the paper is that it blurs the distinction between prior work and the new contribution. For example, the subsection on Split Linearized Bregman Iteration in the "Methodology" section does not contain anything new compared to [1], and this is not clear enough to the reader. Also, not enough credit is given to [2] for the "Differential of Inclusion of Inverse Scale Space" subsection. I maintain that there needs to be a separate "Background" section for this, and that it should be made absolutely clear what is new in the "Methodology" section. It feels like this distinction is obfuscated in the writing.

* Given that the authors propose Split-LBI as a new optimizer that can be compared to others (e.g., Adam), one issue is that there doesn't seem to be any search done over the hyperparameters of each optimizer, including the learning rate and amount of weight decay. For example, Adam is only run with learning rate 1e-3 and SGD is run with 0.1; in addition, the weight decay (where used) is only set to 1e-4. Thus, it is not clear how meaningful these comparisons are. Also, in Table 1, the CIFAR-10 test accuracies are fairly low at ~90%, while modern models such as Wide ResNets can achieve ~95%.

* The newly-written conclusion is still incorrect, stating again that Split LBI achieves SOTA performance on ImageNet. Also, if  "with better interpretability than SGD" refers to the qualitative comparison of the learned filters, I think this conclusion is a bit too strong, because I don't think the difference is visible enough to aid interpretability.

* Minor point: On further inspection, the legend in the left-side plots in Figure 2 does not match the labels of the visualizations on the right. There is no yellow training curve in Figure 2, despite the assertion in the rebuttal.


[1] Huang et al., Split LBI: An iterative regularization path with structural sparsity. NeurIPS 2016.
[2] Osher et al., "Sparse recovery via differential inclusions." Applied and Computational Harmonic Analysis, 2016.


I maintain my score of weak reject, but am not totally opposed to it being accepted, because it provides a way to find sparse networks more efficiently.

**Experience Assessment:**

I have read many papers in this area.

**Review Assessment: Checking Correctness Of Derivations And Theory:**

I assessed the sensibility of the derivations and theory.

**Review Assessment: Checking Correctness Of Experiments:**

I assessed the sensibility of the experiments.

**Review Assessment: Thoroughness In Paper Reading:**

I read the paper at least twice and used my best judgement in assessing the paper.

---

> ### Author Response · Authors · 2019-11-15
> **[Part 1] For Reviewer#4**
>
> Q1: The Split LBI method is presented as a novel contribution (in the abstract: "we propose a new approach based on differential inclusions of inverse scale spaces ..."), but it was already described in detail in a paper that they cite ([3]) published at NeurIPS 2016. The only theoretical contribution of this paper is section 3: Global convergence of Split LBI.
>
> A1: Our full abstract says that the proposed algorithm in this paper is to address the neural network training by incorporating both overparameterization and structural sparsity. It is quite different from that in [3] and relevant references therein. The SplitLBI [3], together with LBI [4] only deal with the linear model whose proved properties do not hold for nonconvex neural network training, while this paper extends to train overparameterized neural networks with structural sparsity, whose practicality and convergence have never been studied. It is nontrivial to make such an extension. For example, (i) we have to introduced the sparsity-enforcement penalty for convolutional filters; and the empirical loss is not squared loss function (ii) to accelerate the training procedure, we extend the SplitLBI to have stochastic training (batch Split LBI) with momentum and weight decay and thus can be employed to efficiently training the neural networks. In particular, we have established the global convergence property for the highly nonconvex training of deep neural network, by proving the Kurdyka-Lojasiewicz (KL) property for such problems.
>
>
> Q2: The claim that "Split LBI demonstrates SOTA performance in large scale training on ImageNet" (from the abstract) is not correct, and needs to be qualified. This paper reports 70.55/89.56% (top 1 / top 5) accuracy; as far as I am aware, the current SOTA on ImageNet is [6], which achieves 86.4/98.0% (top 1 / top 5) accuracy. I think it would be better for this paper to argue that it achieves comparable performance to baselines with a particular architecture and training regime.
>
> A2: Thanks for this point. Under the same architecture (Resnet-18), our SplitLBI outperforms existing algorithms. We have updated this claim in our paper, marked in red.
>
> Q3: The structure and presentation of the paper could be improved in several ways, outlined as follows.
>
> (Q3-1): Most of the Methodology section discusses the Split LBI method from prior work. I would encourage the authors to split the Methodology section into a separate Background section for Split LBI, followed by a new section specifically about applying Split-LBI to convolutional and fully-connected layers in neural networks.
> (A3-1):  Thanks for your suggestion. In the literature, the SplitLBI was proposed and studied with linear models. In our paper, we have extended it to neural training with general loss function and sparsity regularization, including group sparsity for convolutional layers. Such algorithms for neural training have not been studied in literature and SplitLBI seems thus not familiar to the society. A paragraph in the introduction and some detailed comments in the methodology both highlight the related work and our new extension. If page limit allows, we may try to restructure these comments to a separate background section.
>
> (Q3-2): The writing is missing some details and explanations that would be very helpful for readers. For example, it should clearly state that the dimension of \Gamma is the same as the dimension of the weights W.
> (A3-2): Thanks for your advice! We have accordingly updated the manuscript to make it more readable.
>
> (Q3-3): It would also be good to expand the explanation about why SplitISS avoids the parameter correlation problem, and what it means for \Gamma to have an orthogonal design?
> (A3-3): The orthogonal design of $\Gamma$ is because that the Hessian of augmented loss with respect to $\Gamma$ is orthogonal, therefore the correlation between features can be reduced. A detailed study for linear models has been given in [7] by showing the irrepresentable condition of SplitLBI is easier to meet than that of generalized Lasso, due to the disentanglement between the true features/parameters and nulls. We have mentioned it after Eq (3), marked by red on page 4.
>
> (Q3-4): The figures are too small to be readable without a lot of zooming.
> (A3-4): We have tried our best to make figures as big as possible to make it clearer. However, due to the space limit, it is difficult to further enlarge the figures. We have enlarged the figures in the appendix.

---

> ### Author Response · Authors · 2019-11-15
> **[Part 2] for Reviewer #4**
>
> (Q3-5): The paper ends abruptly, with no conclusion.
>
> (A3-5): Thanks for your advice! We have added a conclusion section. “In this paper, a parsimonious deep learning method is proposed based on differential inclusions of inverse scale spaces. Implemented by a variable splitting scheme, such a dynamics system can exploit over-parameterized models and structural sparsity simultaneously. Besides, its simple discretization, i.e., the SplitLBI, has a proven global convergence and hence can be employed to train deep networks. We have experimentally shown that it can achieve the state-of-the-art performance on many datasets including ImageNet-2012, with better interpretability than SGD. What's more, equipped with early stopping, such a structural sparsity can unveil the "winning tickets" -- the architecture of sub-networks which after re-training can achieve comparable and even better accuracy than original dense networks.”
>
>
> (Q3-6): The appendix contains some useful material, but much of it is not referenced from the main paper. I think some parts of the appendix could be moved to the main paper, for example the comparison of computational and memory costs between Split LBI and SGD.
>
> (A3-6): The main contribution of our methods is (i) global convergence (ii) directly training network via SplitLBI (iii) structural sparsity for better interpretability and architecture of sub-network, of which the results were shown in experimental section (4.1-4.4). In contrast, the computational cost for training time is an aspect of SplitLBI however does not belong to the main contribution of our method we want to highlight, hence due to the space limit, we leave such a part in the appendix.
>
> Q4: What the purpose of the comparisons between optimizers in Table 1 is.
>
> A4: In section 3, we have proved the global convergence of SplitLBI. In section 4, we proposed the mini-batch SplitLBI, ensure its ability to train neural network. Table 1 is a validation that our SplitLBI can be employed to train deep neural network and achieves comparable or even better results than other optimization algorithms.
>
> Q5: Regarding experiments, it is not clear which experiments the authors actually performed, for which they took the results from previously published papers, and sometimes where the results come from at all.
> A5: Actually, it makes sense to directly compare Split LBI with other algorithms in Tab. 1. We answer each question specifically
>
> (Q5-1) The motivation given in the abstract and introduction is to learn sparse networks online during optimization; it does not propose Split LBI as a new optimizer to compete with Adam.
> (A5-1) As the listed in the first contribution, we have “SplitLBI, as an extension of SGD, is applied to deep learning by exploring both over-parameterized models and structural sparsity in the inverse scale space.” For this perspective, we still can term SplitLBI as an optimization algorithm extending from SGD. Thus, it makes sense to compare it with other optimizers.
>
> (Q5-2) Couldn't one use the Adam update rule to optimize the weights in Split-LBI?
> (A5-2). Yes. Indeed, the SplitLBI can incorporate momentum and other variants of gradient descent method like Adam but the theoretical properties established here may not hold. It’s a good future direction.
>
> (Q5-3) it makes sense to compare Split LBI to standard training setups that do not enforce any sparsity, as well as to setups that use L1 and L2 regularization.
> (A5-3). Empirically, the SGD with L2 regularization (weight decay) can help generalization. We have compared with SGD without any regularization, as shown in the first row of last column in Table 1, i.e., (Naïve). As shown, the Naïve version performs worse than L2 regularization on both ImageNet and Cifar10 (Naïve v.s. L2 regularization): Cifar10 89.44 v.s. 90.31; ImageNet (top 1) 66.98 v.s. 69.76; ImageNet (top 5): 86.97 v.s. 89.18.
>
> (Q5-4) many rows in Table 1 are missing data (e.g., the variants of Adam are only run for CIFAR-10).
> (A5-4) On ImageNet-2012, we tried implementations of Adam variants including Radam, Adabound, Adagrad and Amsgrad, which failed to successfully train the network. Besides, we can’t find in  previous literature any reported results which are relevant to using Adam or its variants to train ImageNet. Thus, we have to leave these rows as empty in Tab. 1. We fill the results of Adam variants on MNIST.

---

> ### Author Response · Authors · 2019-11-15
> **[Part 3] for Reviewer #4**
>
> Q6: Regarding experiments, it is not clear which experiments the authors actually performed, for which they took the results from previously published papers, and sometimes where the results come from at all.
>
> (Q6-1): In Table 1, there is an asterix next to SGD-Mom-Wd that the authors say indicates "results from the official pytorch website." That would imply that the rest of the experiments were done by the authors (and the authors say in the caption "we use the official pytorch codes to run the competitors"). However, Table 2 (found in the Appendix, page 20), contains identical numbers and a sign # that, according to the authors, indicates results of their own experiments. That would mean that of all the SGD and Adam experiments shown in the table, the authors only performed SGD-naive and Adam-naive. Where do the other numbers come from? What does "official pytorch code to run the competitors" mean? Where is that code from?
> (A6-1): In our Table 1, we run the experiments by ourselves except for SGD Mom-Wd on ImageNet, of which the results are cited from https://pytorch.org/docs/stable/torchvision/models.html. We have changed the caption and clarified this issue. Regarding to Table 2, we have a typo as Table 1 in our original submission. We are very sorry for the misunderstanding, and we have updated the correct version of Table 2 in our new manuscript.
>
> (Q6-2): Figure 4 contains baselines and SplitLBI results. Where do the numbers for the baselines come from? The caption mentions another paper, [2], but I did not find the source of the numbers in that paper. The caption seems to point to Table 9a in [2], but that table does not deal with Network Slimming, Soft-Filter Pruning, Scratch B, or Scratch-E. Additionally, Table 9a of [2] only contains results for VGG-16 and ResNet-50. Where do the baselines for ResNet-56 (Figure 4b) come from?
> (A6-2). Thank you for the comment, we should state more clearly. Some results are not shown in the paper [2], but the author released the source code
> https://github.com/Eric-mingjie/rethinking-network-pruning
> and thus, these results can be re-produced. We have clarified them in the new version.
>
> Q7: How is the proximal objective in Eq. 5 optimized? That is, how do you compute the argmin?
> A7: Thanks. Eq.5 has an explicit solution: Prox(V) = max( 0, 1-(1/ ||V||) ) * V.
>
> Q8: Figure 2 shows results for SLBI-1 and SLBI-10, but no discussion of what SLBI-1 and SLBI-10 mean. Also, regarding Fig.2, the authors claim that "Filters learned by ImageNet prefer non-semantic texture rather than shape and color." How did the authors come to this conclusion? I looked carefully at the filter visualizations, and I cannot see a clear difference between the filters learned by Split LBI and SGD.
> A8:  The SLBI-1 and SLBI-10 means the SplitLBI with kappa = 1 and =10, respectively. We have updated in the new version. For Figure 2, we ranked the filters from top left to bottom right according to their norm. Note that the first row of SLBI-1 and SLBI-10 (the filters with the largest magnitude) mainly contains the information of background, rather than shape and color, which is in accordance with recent study [5]. In contrast, the filters in the first row of SGD have some including the shape information.
>
> Q9: The computation time comparison in Table 11 (Appendix E) is a bit strange, because it shows that Adam takes 2x as long as SGD, which does not align with my experience; in practice, the wall-clock time is nearly identical between Adam and SGD. It would be good to provide more details about how the time was measured. Also, does the memory comparison measure only the memory used for model parameters (W and \Gamma), or also activation memory? Shouldn't Split LBI use 2x the memory of SGD (if measuring only the weights)?
>
> A9: Thanks for your comment. The original reported computational time only includes mean time for updating the weights. For more precise comparison, we take inference time, gradient calculation time and weights update time into consideration. New results are reported in the revised version. These three methods are similar in terms of this metric.
>
> The memory cost composes of two parts: (i) activation (ii) weight parameters. Compared with weight parameters, the activation is much more costly and is the same for SLBI, SGD and Adam since they share the same model architecture. Hence, the SLBI only costs a bit more for GPU memory than SGD due to the extra need for \Gamma.

---

> ### Author Response · Authors · 2019-11-15
> **[Part 4] for Reviewer #4**
>
>
> Q10: In Figure 1, it looks like the initial magnitude of the filters is larger for SGD compared to Split LBI. Are the weights initialized in the same way? Also, why is the setup of the MNIST experiment in Fig.1 different from the setup in Table 1 (e.g., learning rate decay every 40 epochs vs every 30 epochs)? In addition, it looks like the first learning rate decay causes the filter weight magnitudes to flatten out and stay constant.
>
> A10:  We use the same initialization, while the starting point in the x-axis is the first epoch. In other words, we plotted the curve from the training after the first epoch. In this figure we use \kappa=10, for sufficient growth of \Gamma, we choose to use a longer one as 40 epochs. Empirically, we found that lr decay every 30 epochs has similar experimental result with every 40 epochs. To make a unifying lr decay strategy, we also give one 30 epochs figure in the revised version and we found the similar results as before. We have updated in the new version.
>
> Q11: What is the learning rate schedule used for the runs in Figure 3? It looks like the lr decays at epoch 80 and 120, but this is only mentioned in table captions in the appendix. This should be stated in the main paper. Also, why is this a different training setup from that used for Table 1?
>
> A11 : In Table 1 we cite the Res18 ImageNet results trained with SGD from  https://pytorch.org/docs/stable/torchvision/models.html, of which the learning rate scheme follows from [1] that the initial value is 0.1 and decay for every 30 epochs. For fair comparison we utilize the same policy as Table 1.
> Compared to Table 1 which employed SplitLBI for training dense parameters W, we in Figure 3 focus on training the sparse network in which the architecture can be determined by the support set of \Gamma. For such a sparsity estimator \Gamma, the SplitLBI with larger kappa and nu iterates slowly in terms of selecting non-zeros variable. Therefore, the lr decay with earlier epochs will result in even slower selection of non-zero variables. Hence, to overcome such a problem, here we adopt a longer training time at a learning rate: decay at 80 and 120 epochs.
>
> Q12: I also noted that the authors do not intend to make the code public upon publication of the paper. On page 6, they state that "source codes will be released upon requests." At present, the preferred path is to make the code public upon publication of the paper.
> A12: We have released the source code in the submission system: https://anonymous.4open.science/repository/d22bbbc8-50d5-4e60-b2e8-4ded4e93db63/Split_LBI_code
>
> Q13: In the caption of Figure 3, it says "The results are repeated for 5 times. Shaded area indicates the variance; and in each round, we keep the exactly same initialization for each model." What is different between the 5 runs if the initialization is the same?
> A13: This sentence means that we run for \kappa=1,2,5,10 all for 5 rounds. For each kappa, different rounds have different initializations, while the same set of initializations is the same for different kappa.  We revise this point and make it clearer.
>
> Q14: There are too many different colors used in Figures 1 and 2. Since the purple, green, and black boxes are important to see for figures 1 and 2, it is confusing to have to deal with additional blue, pink, and yellow boxes around every three.
> A14: In Figure 1, color boxes correspond to epochs. In Figure 2, The blue, pink and yellow boxes correspond to training curves of the same color type in the left figure, with each training curve representing the corresponding method. Here color matches the training method.
>
>
> Reference
> 1. Kaiming He, Xiangyu Zhang, Shaoqing Ren, and Jian Sun. Deep residual learning for image
> recognition.
> 2. Liu et al., Rethinking the value of network pruning. ICLR 2019.
> 3 Osher, Stanley, et al. "Sparse recovery via differential inclusions." Applied and Computational Harmonic Analysis 41.2 (2016): 436-469.
> 4 Huang, Chendi, et al. "Split lbi: An iterative regularization path with structural sparsity." Advances In Neural Information Processing Systems. 2016.
> 5 Geirhos, Robert, et al. "ImageNet-trained CNNs are biased towards texture; increasing shape bias improves accuracy and robustness." ICLR 2019. arXiv preprint arXiv:1811.12231 (2018).
> 6 Touvron et al., Fixing the train-test resolution discrepancy. https://arxiv.org/abs/1906.06423.
> 7 Huang, Chendi et al. "Boosting with structural sparsity: a differential inclusion approach." Applied and Computational Harmonic Analysis. arXiv preprint arXiv:1810.03608 (2018).

---

### Author Response · Authors · 2019-11-15
**Summary of Changes**

We thank all the reviewers for their insightful and constructive comments. According to these suggestions, we have revised our submission and major changes are summarized as follows.

1， In abstract, we altered the claim  about ImageNet performance to “ achieve comparable and even better performance than the other training algorithms on ResNet-18” as pointed out by review#4.
2， At the bottom of Page 2, we altered the claim about ImageNet performance in Contribution to “comparable and even better performance than the other training algorithms on ResNet-18 ” as pointed out by review#4.
3， In section 2 middle of Page 3, we added some explanation for variables  as required by review #2.
4，In section 2, line 3 of Page 4, we added explanation for connection between W and \Gamma.
5， In caption of Table 1, we clarified the source of results as required by review#4.
6，In caption of Figure 2, we explained the meaning of (SLBI-1) and (SLBI-10) as suggested by review#3.
7，In section 4.3,the first paragraph and the third paragraph we added  purpose of ablation study and explanations for Figure3 and Figure4 as suggested by review#4.
8， We added Figure 3 about training loss and training accuracy for training Cifar10 as required by review#3.
9，We added learning rate schedule for Figure3 and Figure4 as required by review#4.
10，We added learning rate schedule for Figure3 and Figure4 as required by review#4.
11，In caption of Figure 5 we added completed source of results as required by review#4.
12 ， We added Conclusion as section 5 in Page 10 as required by review#2 and review#4.
13 ， In Appendix C, we altered the Table 6 by changing the time to be the whole time including inference, gradient calculation and weights update as suggested by review#4.

---

### Decision · Program_Chairs · 2019-12-19

**Decision:**

Reject

**Comment:**

This paper investigates an existing method for fitting sparse neural networks, and provides a novel proof of global convergence.  Overall, this seems like a sensible, if marginal, contribution.  However, there were serious red flags regarding the care which which the scholarship was done which make me deem the current submission unsuitable for publication.  In particular, two points raised by R4, which were not addressed even after the rebuttal:

1) "One important issue with the paper is that it blurs the distinction between prior work and the new contribution. For example, the subsection on Split Linearized Bregman Iteration in the "Methodology" section does not contain anything new compared to [1], and this is not clear enough to the reader."

2) "The newly-written conclusion is still incorrect, stating again that Split LBI achieves SOTA performance on ImageNet."

I believe that R3's high score is due to not noticing these unsupported or misleading claims.